Resource

# Highly active rubiscos discovered by systematic interrogation of natural sequence diversity

Dan Davidi[1,†,¶], Melina Shamshoum[1,¶], Zhijun Guo[2,¶], Yinon M Bar-On[1], Noam Prywes[3], Aia Oz[4,5], Jagoda Jablonska[6], Avi Flamholz[3] [ID], David G Wernick[1,‡], Niv Antonovsky[1,§], Benoit de Pins[1], Lior Shachar[1], Dina Hochhauser[7], Yoav Peleg[8], Shira Albeck[8], Itai Sharon[4,5], Oliver Mueller-Cajar[2] & Ron Milo[1,*] [ID]

## Abstract

$CO_2$ is converted into biomass almost solely by the enzyme rubisco. The poor carboxylation properties of plant rubiscos have led to efforts that made it the most kinetically characterized enzyme, yet these studies focused on < 5% of its natural diversity. Here, we searched for fast-carboxylating variants by systematically mining genomic and metagenomic data. Approximately 33,000 unique rubisco sequences were identified and clustered into ≈ 1,000 similarity groups. We then synthesized, purified, and biochemically tested the carboxylation rates of 143 representatives, spanning all clusters of form-II and form-II/III rubiscos. Most variants (> 100) were active *in vitro*, with the fastest having a turnover number of $22 \pm 1$ s$^{-1}$—sixfold faster than the median plant rubisco and nearly twofold faster than the fastest measured rubisco to date. Unlike rubiscos from plants and cyanobacteria, the fastest variants discovered here are homodimers and exhibit a much simpler folding and activation kinetics. Our pipeline can be utilized to explore the kinetic space of other enzymes of interest, allowing us to get a better view of the biosynthetic potential of the biosphere.

**Keywords** carbon fixation; carboxylation rate; enhanced photosynthesis; metagenomic survey; ribulose-1,5-bisphosphate carboxylase/oxygenase
**Subject Categories** Metabolism; Plant Biology
**The EMBO Journal (2020) 39: e104081**

## Introduction

Increasing the rate of carbon fixation is a pressing challenge toward more sustainable food and energy production. For crops under human cultivation, where water and fertilizers are not limited, carbon fixation is often the limiting factor (Andrews & Whitney, 2003; Parry *et al*, 2007; Bar-Even *et al*, 2010; Raines, 2011). Carbon fixation is itself limited, at least in some cases, by the rate of the carboxylation step, which is catalyzed by ribulose-1,5-bisphosphate (RuBP) carboxylase/oxygenase, commonly known as rubisco (Jensen, 2000). Rubisco is found in nature in at least three distinct forms (Erb & Zarzycki, 2018). Form-I rubiscos are employed by plants, algae, and photosynthetic bacteria and are composed of large (L) and small (S) subunits in an $L_8S_8$ stoichiometry (Tabita *et al*, 2007; Satagopan *et al*, 2014). Form-II rubiscos are found in diverse types of bacteria and dinoflagellates and are simpler, comprising only a large subunit and commonly assuming an $L_2$ or $L_6$ oligomer structure (Whitney *et al*, 2011). Form-III rubiscos have been identified in both bacteria and archaea and typically form $L_2$ or $L_{10}$ complexes (Tabita *et al*, 2008). As indicated by recent metagenomic data, such coarse-grained segmentation is simplistic, as several other rubisco forms exist, e.g., form-II/III, which are found in both bacteria and archaea (Jaffe *et al*, 2019). Notably, there is a family of structurally related rubisco-like proteins that are denoted form-IV rubiscos despite not catalyzing RuBP, $CO_2$, or $O_2$ (Tabita *et al*, 2007).

Given its central role in the biosphere, it is unsurprising that rubisco is one of the most thoroughly studied enzymes (Jeske *et al*, 2019). Over the last five decades, turnover numbers ($k_{cat}$ values) of rubisco from over 200 organisms have been reported. This exceeds

1 Department of Plant and Environmental Sciences, Weizmann Institute of Science, Rehovot, Israel
2 School of Biological Sciences, Nanyang Technological University, Singapore, Singapore
3 Department of Molecular and Cell Biology, University of California, Berkeley, CA, USA
4 Migal Galilee Research Institute, Kiryat Shmona, Israel
5 Tel Hai College, Upper Galilee, Israel
6 Department of Biomolecular Sciences, Weizmann Institute of Science, Rehovot, Israel
7 Department of Molecular Genetics, Weizmann Institute of Science, Rehovot, Israel
8 Department of Life Sciences Core Facilities, Weizmann Institute of Science, Rehovot, Israel
*Corresponding author. Tel: +972505714697; E-mail: ron.milo@weizmann.ac.il
¶These authors contributed equally to this work
†Present address: Department of Genetics, Harvard Medical School, Boston, MA, USA
‡Present address: BASF Enzymes LLC, San Diego, CA, USA
§Present address: Laboratory of Genetically Encoded Small Molecules, The Rockefeller University, New York, NY, USA

by a factor of two the aggregated number of $k_{cat}$ measurements for the next three best-studied enzymes (i.e., carbonic anhydrase, beta-lactamase, and DHFR (Jeske *et al*, 2019; Flamholz *et al*, 2019)). The assembled $k_{cat}$ values suggest that rubisco is a kinetically constrained enzyme. First, the median measured $k_{cat}$ value is 3.3 s$^{-1}$, threefold slower than the median enzyme (Bar-Even *et al*, 2011; Davidi *et al*, 2018). Second, despite being so overrepresented in kinetic studies, rubisco's $k_{cat}$ values span the smallest dynamic range across all catalytically characterized enzymes (Flamholz *et al*, 2019). Third, multiple efforts to improve the rate of rubisco carboxylation have made limited progress (Mueller-Cajar *et al*, 2007; Gomez-Fernandez *et al*, 2018; Wilson *et al*, 2018). Cases where some increase in the net carboxylation capacity of rubisco was achieved (via directed evolution) include only form-I and form-III isoforms (Wilson *et al*, 2016; Zhou & Whitney, 2019).

Indeed, it has been suggested that the catalytic properties of this enzyme may already be optimized and thus cannot be straightforwardly improved (Tcherkez *et al*, 2006; Savir *et al*, 2010; Bathellier *et al*, 2018). Specifically, rubisco can use $O_2$ as a substrate instead of $CO_2$ to catalyze an oxygenation reaction. A tradeoff between greater $CO_2:O_2$ specificity ($S_{C/O}$) and a greater carboxylation rate was previously reported (Tcherkez *et al*, 2006; Savir *et al*, 2010). In this tradeoff, the enzyme might have reached what was discussed as a fitness optimum. However, this perception is based on kinetic measurements that cover only a tiny fraction of rubisco's natural diversity. In a recent analysis, we showed that measurements of rubisco homologs from a set of divergent organisms have called the robustness of the "optimality hypothesis" into question (Flamholz *et al*, 2019). If rubisco's carboxylation kinetics are not tightly constrained, we might expect to find fast-carboxylating variants in nature. Prokaryotic rubiscos are particularly promising given their immense ecological and sequence diversity (Tabita, 1999; Badger & Bek, 2008; Witte *et al*, 2010) and the meager data available on their kinetics.

This work, therefore, aims at a systematic survey of the kinetic space of natural prokaryotic rubiscos in search of catalytic outliers in terms of $CO_2$ fixation speed ($k_{cat}$ for carboxylation). For this purpose, we developed a pipeline for bioinformatic mining, gene synthesis, protein purification, and biochemical characterization of rubisco homologs (Fig 1). We then applied it to investigate a representative diversity-spanning set of 143 rubisco variants, asking whether evolution gave rise to catalysts, that under standard biochemical conditions, exhibit higher $CO_2$ fixation kinetics.

# Results

### Mapping the sequence space of wild rubiscos

To map the natural diversity of rubisco sequences, we performed an exhaustive search for rubisco homologs across the major genomic and metagenomic public databases using rubisco's large subunit from *Rhodospirillum rubrum* as a bait (in April 2017; see Materials and Methods). After filtering truncated and ambiguous sequences, $\approx$ 40,000 non-redundant homologs were identified from all three domains of life. To identify the form of each variant, a set of manually curated variants was mapped to a phylogenetic tree and rubisco-like proteins were removed as they lack carboxylation

activity, leaving $\approx$ 33,000 non-redundant variants (see Materials and Methods for a detailed description of the pipeline). While rubisco enzymes that do not share sequence homology to currently identified variants are not detectable by this pipeline, so far, no evolutionarily distinct rubiscos were found (Böhnke & Perner, 2019). We, therefore, denote the set of $\approx$ 33,000 sequences as the sequence space of wild rubisco's large subunit.

To best represent the sequence space, rubisco variants were clustered at 90% protein sequence identity into 833 groups. Approximately 32,000 sequences spanning 460 clusters were identified as form-I. We also found 336 form-II variants spanning 119 clusters, 815 form-III rubiscos spanning 231 clusters, and 37 form-II/III spanning 23 clusters (see Materials and Methods). Notably, while 97% of the sequenced rubiscos are form-I, they account for only $\approx$ 50% of the sequence diversity of rubisco's large subunit. It is likely that future metagenomic efforts will uncover novel rubisco sequences that shall further expand the contribution of form-II, form-III, and form-II/III rubiscos to the global sequence diversity of this enzyme.

In accordance with their ubiquity in sequence databases, form-I rubiscos are also highly overrepresented in kinetic studies. Ninety-five percent of reported rubisco kinetic values are of form-I (80% are from plants), with only a handful of studies accounting for other forms (Flamholz *et al*, 2019; Jeske *et al*, 2019). Altogether, kinetic measurements cover merely 33 of 833 diversity clusters, highlighting how sparsely characterized this pivotal enzyme.

### Expanding the space of $k_{cat}$ characterized rubiscos by fourfold

Here, we explore 143 cluster representative rubiscos, 140 of which are from untested clusters, thereby increasing the coverage of explored sequence diversity by fourfold (Fig 2). As many of the sequences we identified come from unculturable organisms or metagenomic assemblies, we chose to purify recombinantly expressed rubiscos for kinetic characterization. Recombinant production of form-I and form-III rubiscos has often proven challenging. The folding and assembly of plant form-I rubisco into an $L_8S_8$ holoenzyme is a multistep process that relies on several supporting chaperones (Aigner *et al*, 2017). Chaperone compatibility issues between even highly similar plant rubiscos make a systematic exploration of form-I rubiscos infeasible at the current stage (Sharwood *et al*, 2016; Bracher *et al*, 2017). Form-III rubiscos were excluded from our study based on a pilot experiment of 18 variants from distinct sequence-similarity clusters. Out of 18 tested variants, only two variants had carboxylation rates above background levels.

Form-II variants are often successfully overexpressed in model organisms like *Escherichia coli* and *Nicotiana tabacum* (Pierce *et al*, 1989; Whitney & Andrews, 2001, 2003; Mueller-Cajar *et al*, 2007; Antonovsky *et al*, 2016). They are structurally simpler than form-I, and the few variants measured so far exhibit relatively fast $k_{cat}$ values (Flamholz *et al*, 2019). Further, form-II sequences are very diverse, spanning more diversity clusters than are currently represented in literature measurements (Fig 2). We also sampled form-II/III diversity, as it represents an interesting case study between form-II and form-III. As shown in Fig 2, our chosen set of 143 diversity clusters covers the entire sequence space of form-II and form-II/III rubiscos at 90% sequence identity.

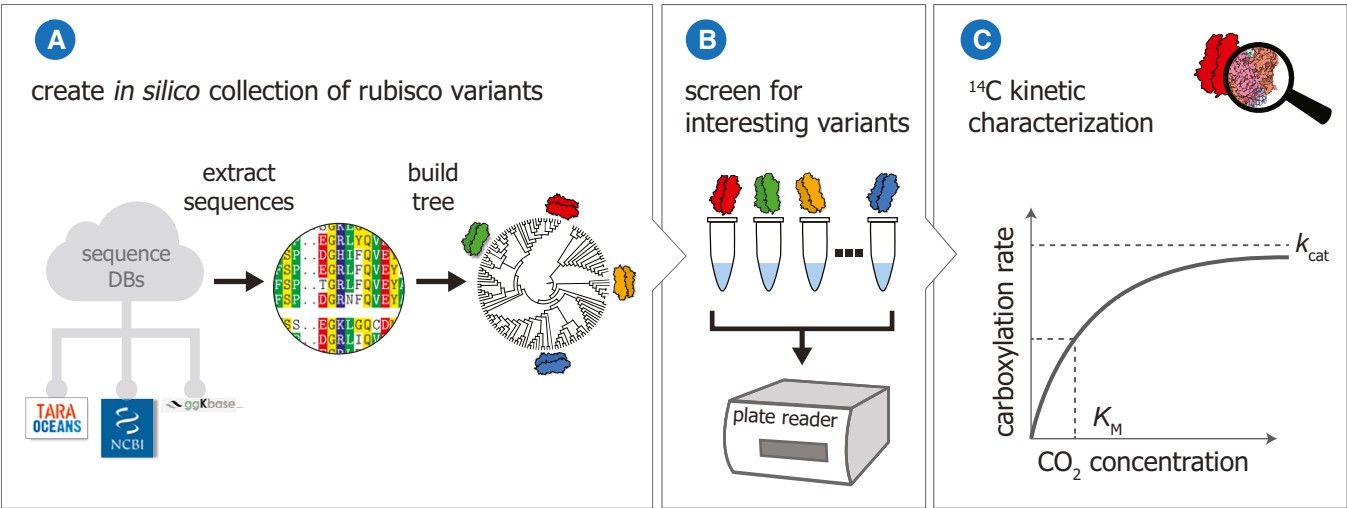

**Figure 1. Schematic of the workflow developed and employed in this study.**

A A computational pipeline to extract rubisco sequences from genomic and metagenomic databases, cluster them based on sequence identity, and select representatives that cover the entire diversity of rubisco variants from nature.
B An experimental pipeline to screen the representative variants for carboxylation activity.
C Catalytic outliers are evaluated using radiometric assays for the determination of accurate $k_{cat}$ and $K_M$ carboxylation values as well as $S_{C/O}$.

## A high-throughput assay for active-site carboxylation rate determination

Rubisco genes were codon-optimized for expression in *E. coli*, synthesized and cloned into a pET28 vector system. Each rubisco was expressed with an N-terminal His-tag followed by a SUMO protease recognition site for post-purification tag cleavage. Cleavage

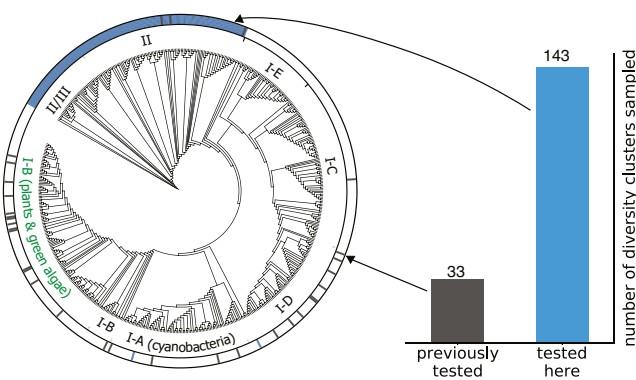

**Figure 2. Systematic exploration of uncharted rubisco representatives outside of the heavily sampled form-I group.**

Of 223 total $k_{cat}$ values reported in the literature, 217 measurements are for form-I, two for form-II, three for form-III, and one for form-II/III (Flamholz et al, 2019). This previously explored diversity coverage is shown on a phylogenetic tree of rubisco homologs at 90% identity, compared to the diversity explored in this study. Each leaf corresponds to a cluster of sequences such that all leaves share < 90% identity. All subtypes of form-I, form-II, and form-II/III rubiscos are annotated. Form-III rubiscos were omitted from the tree for clarity, as they were not biochemically tested here and only three variants from this group have reported kinetics. Form-IV were removed as they apparently lack the carboxylating activity (Tabita et al, 2007) (see Appendix Fig S1 for the full phylogenetic tree).

is scarless and results in purified native rubiscos (see Appendix and Materials and Methods).

The carboxylation rates of each purified rubisco were determined using a spectroscopic coupled assay, where the carboxylation product of RuBP, 3-phosphoglycerate, was coupled to NADH oxidation (Lilley & Walker, 1974; Kubien et al, 2011) (Fig 3A; see also Materials and Methods). Because NADH levels can be measured spectrophotometrically, the decay in NADH absorbance signal reflects the rate of 3-phosphoglycerate production, which is equal to twice the specific activity (two molecules of 3-phosphoglycerate are formed per carboxylation reaction; Fig 3B). A potential caveat to this assay is that if RuBP oxygenation occurs, 3-phosphoglycerate and 2-phosphoglycolate are produced, resulting in an underestimation of the carboxylation rate. To minimize this issue, the spectrophotometric assays were undertaken in solution equilibrated in a nitrogen atmosphere comprising 4% $CO_2$ (100 times atmospheric conditions) and 0.2% $O_2$ (1/100 of atmospheric conditions) in a gas-controlled plate reader at 30°C (see Materials and Methods). These conditions ensured that rubiscos are fully carbamylated ($CO_2$ activated) before adding RuBP to initiate catalysis and that CO2 levels were approaching saturation.

In order to quantify the concentration of rubisco active sites, each variant was assayed under 5 concentrations of 2-C-carboxyarabinitol 1,5-bisphosphate (CABP)—a transition state analog and a stoichiometric rubisco inhibitor commonly used for active-site quantification (Kubien et al, 2011). From the relationship between reaction rates and CABP concentrations, we measured the actual amount of rubisco active sites per variant (Fig 3C; see also Materials and Methods). Importantly, the use of CABP allowed us to measure a per-active-site rate. In principle, this approach can also be applied to crude, lysate-based assays, due to the tight and specific binding properties of CABP to rubisco. This can help account for differences in expression levels and activation kinetics, which can be dramatic

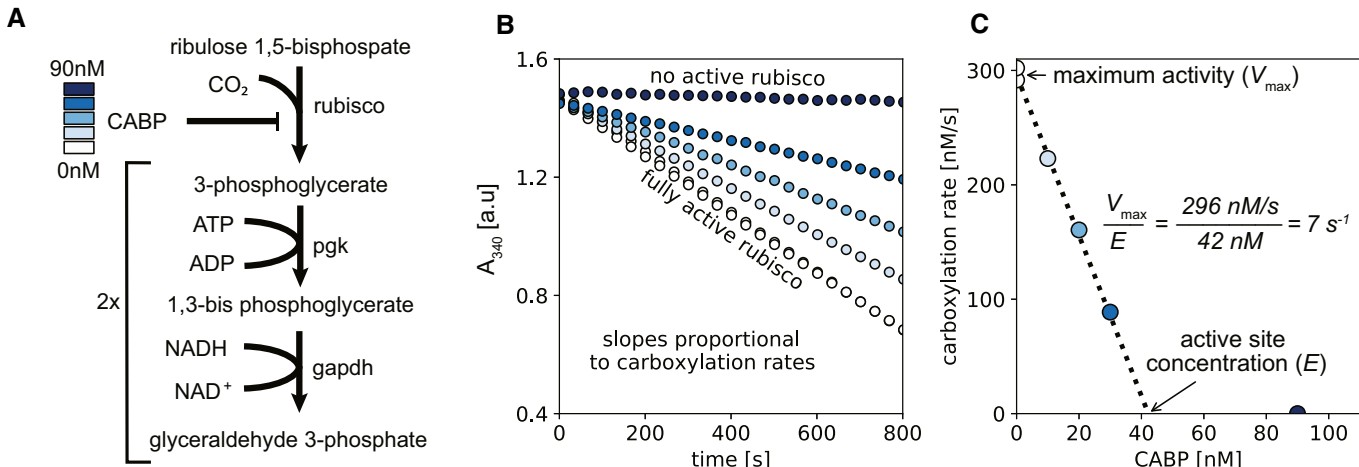

**Figure 3. High-throughput pipeline for measuring the carboxylation rate of rubisco.**

A  Coupling rubisco activity to NADH oxidation is done by two enzymatic steps catalyzed by phosphoglucokinase (pgk) and glyceraldehyde 3-phosphate dehydrogenase (gapdh) (Kubien *et al*, 2011). A gradient of CABP is used to gradually inhibit rubisco activity.

B  NADH oxidation is monitored at 340 nm in a gas-controlled plate reader under 4% $CO_2$ and 0.2% $O_2$ in order to favor carboxylation over oxygenation. The slope of the curves gives the rate of NADH oxidation, which is equal to twice the carboxylation rate; the rate with no CABP was measured in duplicates.

C  Rate of carboxylation (y-axis; slopes from panel B) as a function of CABP concentration (x-axis). The x-intercept gives the concentration of rubisco active sites ([E]) while the y-intercept gives the reaction rate without CABP inhibition ($V_{max}$); thus, the specific activity per active site is given by dividing $V_{max}$ by [E]; dashed line is a least-square linear regression ($r^2 > 0.99$). For this example, rubisco from *R. rubrum* catalyzes ≈ 7 reactions per second.

(see Appendix Figs S2 and S3). However, the pipeline would have to be adjusted to minimize NADH oxidation reactions by native *E. coli* proteins.

### The distribution of carboxylation rates highlights promising variants

Out of 143 synthetic rubisco genes, 105 were successfully assayed for carboxylation activity, i.e., catalyzed ≥ 0.5 reactions per second ($s^{-1}$; Fig 4), a threshold significantly higher than the background noise in our measurements (see Table EV1). The remaining 38 were insoluble in *E. coli* and thus were not biochemically assayed. The median catalytic rate of all active variants was 5.6 $s^{-1}$, similar to the median $k_{cat}$ of plant rubisco (4.7 $s^{-1}$ when corrected to 30°C by assuming a $Q_{10}$ value of 2.2 (Cen & Sage, 2005)). The form-II rubisco from *R. rubrum* was used as a reference, since it often serves as a standard in kinetic assays of rubisco. *R. rubrum* has a median reported $k_{cat}$ of ≈ 7 $s^{-1}$ at 25°C, similar to the value measured here (6.6 $s^{-1}$; Table 1 & Appendix Fig S6). Importantly, as not all variants were tested on the same day, *R. rubrum* rubisco was always included as an internal reference in each measurement (therefore, its measured rate reflects 49 biological replicates with a standard error of 0.3 $s^{-1}$; see Materials and Methods).

A key aspect of this study was the discovery of substantially faster rubiscos. Indeed, several of the variants tested in our high-throughput screen had carboxylation rates above 10 $s^{-1}$, a rate never before observed for form-II rubiscos (Fig 4).

### Discovery of prokaryotic rubiscos with unprecedented $CO_2$ fixation rates

The carboxylation $k_{cat}$ and $K_M$ properties of a subset of seven promising rubiscos were undertaken using high precision $^{14}CO_2$

fixation assays. These assays were conducted on purified enzymes (see Appendix Fig S4) at 25°C and in the absence of the competitive inhibitor $O_2$ (see Materials and Methods for the full protocol). Our measurements yielded some of the fastest $k_{cat}$ values for rubisco reported to date (Fig 5B), testifying to the power of the presented approach. The fastest rubisco was obtained from a soil member of the *Gallionella* genus (OGS68397.1) with a $k_{cat}$ of 22 ± 1.1 $s^{-1}$. This is sixfold faster than the median plant and almost twofold faster than the currently fastest known rubisco from the cyanobacteria *Synechococcus elongatus* PCC 6301 ($k_{cat}$ = 11.7 ± 0.6 $s^{-1}$; Table 1; Fig 5A). We note that a form-III rubisco from the hyperthermophiles archaea *Archaeoglobus fulgidus* has a $k_{cat}$ of 23 $s^{-1}$ at 83°C (Kreel & Tabita, 2015); however, we found that the enzyme shows very little activity at 25°C, consistent with a $k_{cat}$ of ≈ 0.75 $s^{-1}$ predicted assuming $Q_{10}$ = 2.2 (Cen & Sage, 2005). Like *R. rubrum* rubisco, the fast *Gallionella* sp. rubisco was an $L_2$ dimer, as assessed by analytical gel filtration (Table 1; see also Appendix Figs S4–S6).

The $CO_2$:$O_2$ specificity factors ($S_{C/O}$) of the seven fast rubiscos were also measured to compare the rates of RuBP carboxylation over oxygenation under equal $CO_2$ and $O_2$ levels (see Materials and Methods). Compared to *S. elongatus*, which has a specificity factor of 42.7 ± 2.8, the $L_2$ rubisco from *Gallionella* sp. is fourfold less specific (10.0 ± 0.1). The specificity of all our bacterial rubiscos was significantly lower than *S. elongatus*, which is already more than twofold less specific than the median plant rubisco (plant variants exhibit a specificity factor range of ≈ 80–120 (Flamholz *et al*, 2019)). These measurements thus indicate that in order to support considerable carboxylation flux, the novel rubiscos would have to operate in the context of a carbon concentrating mechanism (CCM) or in otherwise high $CO_2$ environments (Ghoshal & Goyal, 2000; Moroney & Ynalvez, 2007; Iñiguez *et al*, 2020).

The *Gallionella* sp. are iron-oxidizing, chemolithotrophic bacteria that live in low-oxygen conditions. Notably, its rubisco has a $CO_2$

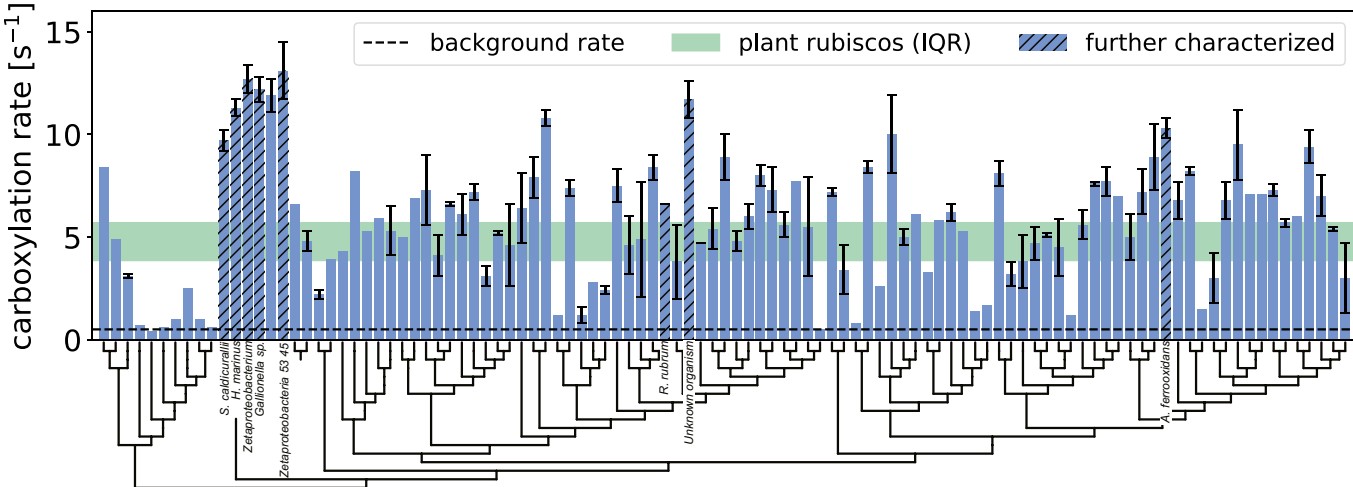

**Figure 4. Measured carboxylation rates for all the variants that were successfully expressed (*N* = 105).**

Rates were determined using a spectrophotometric coupled assay at 30°C (see Materials and Methods). Rubiscos are ordered by phylogeny, as indicated by the dendrogram, and display a weak relationship between sequence similarity and rate of carboxylation, except for one cluster that corresponds to microaerobic bacteria. In green is the interquartile range of $k_{cat}$ values for all reported plant rubiscos from Flamholz *et al* (2019) when corrected for 30°C assuming $Q_{10}$ = 2.2 (Cen & Sage, 2005). Hatched bars represent variants that were analyzed by radiometric analysis, as described below. Bars and error bars correspond to the mean ± standard errors for each variant. The number of replicates for each variant is reported in Table EV1. Values below 0.5 s$^{-1}$ were considered inactive (dashed line). For a full description of the results, see Table EV1.

$K_M$ = 276 ± 6 μM, which is ≈ 30-fold above the soluble $CO_2$ concentration in water at ambient atmospheric conditions (Table 1). This affinity value is similar to that of fast cyanobacterial form-I rubiscos, which have high $CO_2$ $K_M$ values that appear to require an intracellular carbon concentrating mechanism for catalytically efficient $CO_2$ fixation (Badger & Price, 2003). Indeed, the $K_M$ value determined for *S. elongatus* rubisco is 200 μM. The carboxylation efficiency of the *Gallionella* rubisco ($k_{cat}/K_M$) is 80.4 ± 4.5 s$^{-1}$·mM$^{-1}$—40% higher than that of *S. elongatus* and about three-fold lower than the median carboxylation efficiency of plant rubiscos (Table 1; Flamholz *et al*, 2019).

While the catalytic capabilities of enzymes are traditionally evaluated by their $k_{cat}$ or $k_{cat}/K_M$ values, it is important to consider the physiological conditions in which those enzymes are active. Operational *in vivo* carboxylation rates will vary depending on $CO_2$ environments experienced by the enzyme, which are non-saturating if the organism does not possess a CCM or capacity to grow under low $O_2$ or anaerobic conditions. Importantly, the Michaelis–Menten curve for *Gallionella* sp. is above that of *S. elongatus* in all tested $CO_2$ concentrations (Fig 5A), implying that when oxygen is not limiting carboxylation, rubisco from *Gallionella* sp. will catalyze more $CO_2$ fixation reactions compared to *S. elongatus* rubisco under any physiologically relevant $CO_2$ concentration. Plant rubiscos, however, will catalyze more $CO_2$ fixation reactions at $CO_2$ concentrations lower than 50 μM.

Further, four of the seven fastest variants are from bacteria that grow under low-oxygen conditions. Two others come from autotrophic bacteria with flexible metabolisms that encode multiple rubiscos in addition to the form-II variants tested here. Their form-II variants are likely active under anaerobic conditions (Appia-Ayme *et al*, 2006; Esparza *et al*, 2010; Toyoda *et al*, 2018). The final variant comes from an unknown organism with no assembled genome. Similar to the rubisco from *Gallionella* sp., all other fast candidates have $CO_2$ $K_M$ values above 100 μM (Fig 5B).

## Discussion

Rubisco is one of the most thoroughly characterized enzymes, but biochemists have mostly focused on those from plants, whose diversity is very limited. Correlations among rubisco variants characterized so far have led to the hypothesis that faster-carboxylating rubiscos may be found in anaerobic prokaryotes (Tcherkez *et al*, 2006; Flamholz *et al*, 2019). Here, we leveraged advances in DNA sequencing and gene synthesis to interrogate rubisco genes from diverse prokaryotes, which were previously out of reach, as their hosts usually cannot be cultured. Expanding the sequence diversity of catalytically characterized rubiscos by fourfold, our systematic approach was able to reveal seven variants with especially high maximal carboxylation rates—all with $k_{cat}$ values ≥ 10 s$^{-1}$, placing them in the 95% percentile of all reported $k_{cat}$ values to date (Fig 5B). This finding highlights the capability of our pipeline to discover fast rubiscos from nature. More importantly, it hints that the kinetic span of rubisco may not be quite as narrow as suggested by previous literature data.

Our study focused on form-II and form-II/III rubiscos, as they often express well in *E. coli* and are diverse enough to cover a rich kinetic space. All variants with $k_{cat}$ ≥ 10 s$^{-1}$ were form-II rubiscos. A possible explanation could be that form-II rubiscos function as carbon-fixing enzymes in autotrophic organisms, while form-II/III enzymes serve a catabolic role in the salvage of nucleosides (Wrighton *et al*, 2016). Being enzymes of central carbon metabolism, form-II rubiscos may have experienced stronger selection to operate at faster carboxylation rates than form-II/III enzymes. Indeed, central metabolic enzymes have, on average, $k_{cat}$ values 5–10 times faster than those of specialized metabolic enzymes (Bar-Even *et al*, 2011).

Form-IIIs, which are largely of archaeal origin, were excluded from this study as their assembly requirements are not well met by *E. coli*. Yet, exploring the kinetic space of form-III rubiscos is a worthwhile future effort. Their vast sequence diversity alongside

**Table 1.** Values of $k_{cat}$ and $K_M$ for $CO_2$, $S_{C/O}$, and oligomerization states determined for the fastest seven variants from the spectroscopic screen (see also Appendix Figs SS and S6).

| Organism | Form | Oligomer | $k_{cat}$ [s$^{-1}$] | $K_M$ [μM] | $k_{cat}/K_M$ [s$^{-1}$·mM$^{-1}$] | $S_{c/o}$ |
|---|---|---|---|---|---|---|
| *Gallionella* sp. | II | L$_2$ | 22.2 ± 1.1 | 276 ± 6 | 80 ± 4.5 | 10.0 ± 0.1 |
| *Zetaproteobacterium* | II | L$_2$ | 18.2 ± 1.1 | 261 ± 6 | 69 ± 4.4 | 12.5 ± 0.4 |
| *Hydrogenovibrio marinus* | II | L$_2$ | 15.6 ± 0.8 | 162 ± 4 | 96 ± 5.5 | 20.7 ± 2.3 |
| *Sulfurivirga caldicuralii* | II | L$_4$ | 14.3 ± 0.2 | 143 ± 1 | 100 ± 1.6 | 13.5 ± 0.3 |
| *Unknown organism* | II | L$_6$ | 11.8 ± 0.8 | 130 ± 4 | 90 ± 6.8 | 4.8 ± 0.6 |
| *S. elongatus* | I | L$_8$S$_8$ | 11.7 ± 0.6 | 200 ± 4 | 58 ± 3.2 | 42.7 ± 2.8 |
| *Zetaproteobacteria_53_45* | II | L$_2$ | 11.0 ± 0.7 | 284 ± 5 | 38 ± 2.6 | 13.9 ± 0.1 |
| *Acidithiobacillus ferrooxidans* | II | L$_6$ | 9.6 ± 0.7 | 239 ± 5 | 40 ± 3.1 | 16.4 ± 1.3 |
| *R. rubrum* | II | L$_2$ | 6.6 ± 0.3 | 109 ± 2 | 60 ± 3.0 | 12.5 ± 0.6 |
| *Plant (median)* | I | L$_8$S$_8$ | 3.3 ± 1.0 | 14 ± 6 | 230 ± 12 | 94 ± 11 |

Variants were assayed alongside rubiscos from *R. rubrum* and *S. elongatus* as controls. Data are sorted by the inferred carboxylation rate, $k_{cat}$. The kinetic constants $k_{cat}$, $K_M$, and $S_{C/O}$ were determined by radiometric assays (see Materials and Methods). Uncertainties represent standard deviations among technical replicates performed on different days. Values for the median plant rubisco are calculated from Flamholz *et al* (2019).

their alternative biological function may have allowed for alternative evolutionary adaptation in terms of their $CO_2$ and $O_2$ fixation properties. It would be interesting to test whether other heterologous systems and/or alternative codon-usage approaches enable the biochemical characterization of this diverse group (Mignon *et al*, 2018; Satagopan *et al*, 2019).

Here, we tested > 100 rubisco variants, which together cover the space of form-II and form-II/III sequences at a resolution of 90% sequence identity. As can be appreciated from Fig 4, those in close proximity on the phylogenetic tree, reflecting ≈ 90% sequence identity, often exhibit very different rates. Indeed, we calculated the kinetic similarity between all pairs (defined as

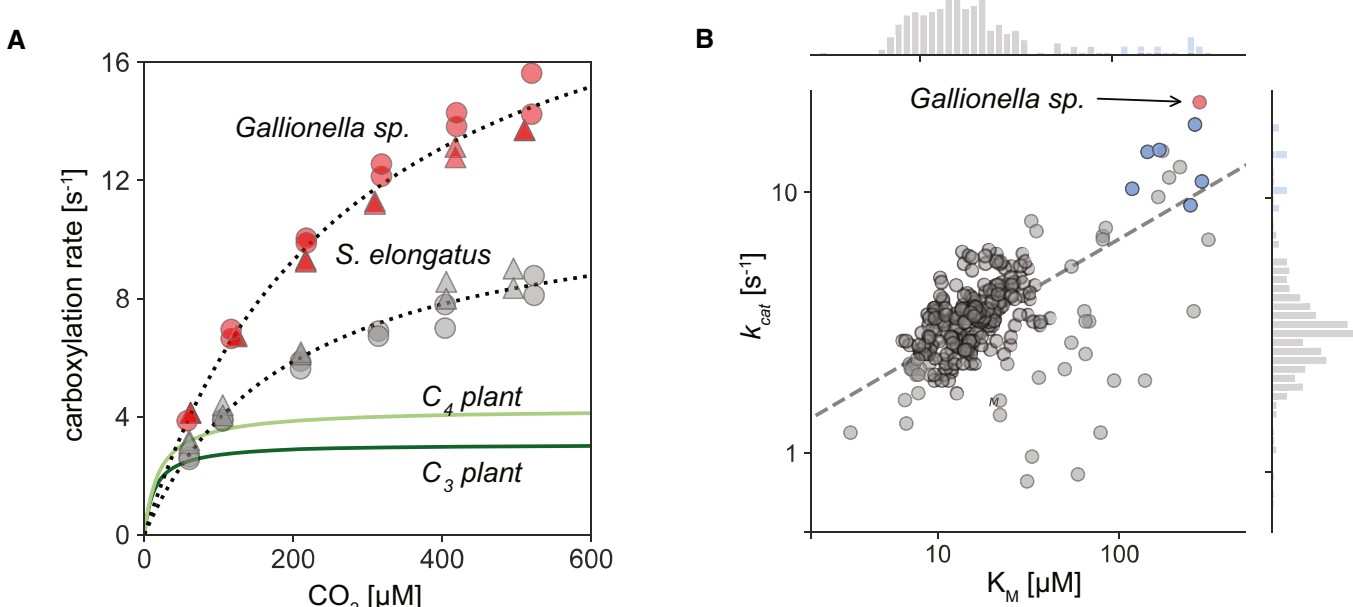

**Figure 5.** Comparing the carboxylation kinetics of *Gallionella* sp. rubisco to reported values from all characterized rubiscos.

A  Michaelis–Menten kinetic plots of rubiscos from *Gallionella* sp. (red; $k_{cat}$ = 22 ± 1.1 s$^{-1}$; $K_M$ = 276 ± 6 μM) and *S. elongatus* (gray; $k_{cat}$ = 11.7 ± 0.6 s$^{-1}$; $K_M$ = 200 ± 4 μM), measured by $^{14}$C labeling in the absence of oxygen at 25°C (see Materials and Methods); variants were measured in two biological repeats (circles and triangles), each in duplicate; dark and light green lines represent the median C$_3$ ($k_{cat}$ = 3.1 s$^{-1}$; $K_M$ = 14 μM) and C$_4$ ($k_{cat}$ = 4.2 s$^{-1}$; $K_M$ = 20 μM) plant rubiscos, respectively.

B  $k_{cat}$ and $K_M$ values for all previously measured rubiscos (Flamholz *et al*, 2019) (gray) and the seven promising variants tested here (red for *Gallionella* sp. and blue for the other six); dashed line indicates the least-square linear regression fit (slope in log scale is 0.4; $r^2$ = 0.44); histograms for $k_{cat}$ and $K_M$ are plotted on parallel axes and clearly show that the variants discovered in our screen are outliers (95[th] percentile) in both $k_{cat}$ and $K_M$ for $CO_2$.

rate$_i$/rate$_j$ for all pairs ($i$, $j$) such that rate$_i$ > rate$_j$ as a function of the percent of protein sequence identity and found the kinetic similarity among sequences with > 80% sequence similarity to be highly similar to the average value calculated across all pairs (1.59 vs. 1.62, respectively). This indicates that (i) rubiscos that are similar in sequence are still mostly kinetically independent and (ii) sampling at a coarser resolution would have lowered the probability of finding fast variants. That being said, one basal clade of relatively fast form-II variants does emerge (Fig 4). It is abundant in microaerophilic iron- and sulfur-oxidizing proteobacteria, with notable representatives from the *Mariprofundus*, *Thiomicrospira*, and *Ferriphaselus* genera (Table EV1), which may serve as a natural starting point for the discovery of additional novel sequences with fast kinetics. Expanding the coverage of the rubisco kinetic space can shed light on as yet unanswered questions regarding its mechanism of action and hypothesized intrinsic catalytic limitations (Tcherkez *et al*, 2006; Savir *et al*, 2010; Studer *et al*, 2014).

Lastly, the mining approach described in this work can be used to identify clusters of related enzymes from which representatives can be chosen to lessen the number of (or more strategically inform which) variants to kinetically evaluate—providing the necessary structure–function appreciation needed to bioengineer an enzyme. The current view of enzyme kinetics is extremely narrow (Davidi & Milo, 2017). Systematic exploration of the natural kinetic space will enable us to reveal the enormous genomic diversity of the biosphere, to understand enzyme sequence–function relationships and to discover superior biocatalysts.

# Materials and Methods

## Mining rubisco sequences from published datasets

To obtain a comprehensive set of rubisco sequences, rubisco from *R. rubrum* (WP_011390153.1) was used as a seed for BLASTp against (i) the non-redundant database from NCBI (Benson *et al*, 2013), (ii) TARA ocean metagenomics (assemblies of 244 samples) (Pesant *et al*, 2015), and (iii) a database of candidate phyla radiation genomes recovered from sediment metagenomes (Brown *et al*, 2015); data were downloaded on April 2017. The BLASTp query was run using an *E*-value threshold of $10^{-5}$. A protein-length filter was applied such that sequences not within the range of 300 < l < 700 amino acids were removed. We further excluded sequences with ambiguous amino acid calls and aberrant sequences (e.g., fusion proteins and false positives), by aligning all variants against *R. rubrum* rubisco and filtering out sequences sharing < 50% alignment coverage (local alignment using BLAST; default parameters). Ultimately, we found 35,413 non-redundant rubisco sequences.

## Rubisco form assignment and removal of rubisco-like proteins

To infer the form of each rubisco homolog, non-redundant sequences were clustered based on protein sequence identity (USEARCH algorithm (Edgar, 2010)) at a 70% identity threshold and cluster centroids were aligned using MAFFT (default parameters) (Katoh & Standley, 2013). The alignment was trimmed such that only positions present in > 95% of the alignment columns were maintained,

with a maximum-likelihood tree generated using RAxML-HPC v8.1.24, based on the REST API implemented on cipres.org (default parameters with rapid bootstrapping) (Stamatakis, 2014). We then mapped a set of manually curated rubiscos with form annotations from (Jaffe *et al*, 2019) onto the 70% identity tree, and each clade of sequences was assigned a form according to the form identity of curated proteins that clustered together with it (all known rubisco forms were identified, i.e., forms Ia, Ib, Ic, Id, Ie, II, II/III, IIIa, IIIb, IIIc, III-like, and IV; see Appendix Fig S1). Clades with an ambiguous phylogenetic affiliation were labeled "unknown", and form-IV rubiscos, which are homologous to rubisco but lack carboxylating activity, were filtered out, leaving us with 33,565 carboxylating rubisco sequences.

## A phylogenetic tree of carboxylating rubiscos at 90% diversity resolution

USEARCH was applied to the set of non-redundant carboxylating rubiscos at a 90% identity threshold. We repeated the same procedure described for generating 70% identity maximum-likelihood tree above to generate a 90% identity maximum-likelihood tree of rubisco variants. We aligned representative sequences from each 90% identity cluster using MAFFT, trimmed the alignment, and built the tree using RAxML-HPC. Data from (Flamholz *et al*, 2019) were used to highlight previously reported variants with characterized kinetic parameters. Matching was performed based on the organism name. Clusters that contained at least one previously reported variant were denoted "previously characterized".

## Gene synthesis

For sampling the sequence space of wild rubiscos, representative sequences from each of the form-II and form-II/III clusters were selected for gene synthesis. Protein sequences were reverse-translated and codon-optimized for expression in *E. coli* using a custom python script (maximum GC content threshold of 60% and Codon Adaptation Index larger than 60%; for codon usage in *E. coli*, we followed Athey *et al* (2017)). Following codon optimization, sequences were synthesized by Twist Bioscience and cloned into a custom overexpression vector. The expression vector used in this study, pET28-14His-bdSumo, was established by cloning of a 14× His-tag followed by a bdSUMO tag cassette described in Frey and Görlich (2014) into the expression vector pET28-TEVH (Peleg & Unger, 2008). Rubisco variants were synthesized into pET28-14xHis-bdSumo by TWIST bioscience and were validated using next-generation sequencing as part of the TWIST bioscience service (see Appendix).

## High-throughput expression and purification

In order to express and purify the synthesized variants, clones were transformed into *E. coli* BL21(DE3) host cells and incubated at 37°C, 250 rpm in 8 ml of LB media supplemented with 50 μg/ml kanamycin; growth was performed in 24-deep-well plates. When cells reached an OD$_{600}$ of 0.8, rubisco expression was induced by adding 0.2 mM IPTG (isopropyl β-D-thiogalactoside) and incubating at 16°C for 16 h. For protein extraction and purification, cells were harvested by centrifugation (15 min; 4,000 *g*; 4°C) and pellets were lysed with BugBuster® ready mix (Millipore) for 25 min at room

temperature; 0.5 ml BugBuster master mix (EMD Millipore) was added to each sample. Crude extracts were then centrifuged for 30 min at 4,000 *g*, 4°C to remove the insoluble fraction. The soluble fraction was then transferred to 96-deep-well plates for His-tag purification. We used a Nickel magnetic bead system (PureProteome™; Millipore), according to the manufacturer's protocol, for washing and binding. Rubisco was eluted by on-bead cleavage of the SUMO tag with bdSENP1 protease. A 150 μl cleavage buffer (20 mM EPPS pH 8.0; 50 mM NaCl; 20 mM MgCl$_2$; 15 mM imidazole) containing bdSENP1 protease (8 μg/ml) was added to each well and incubated for 1 hour on a plate shaker (1,000 rpm; Dlab MX-M Microplate mixer; 25°C). Purified proteins were then separated from the tag-bounded magnetic beads using a magnetic rack and stored at 4°C until kinetic measurement was applied (up to 24 h later). Protein concentrations were measured using a Pierce™ BCA Protein Assay Kit (Thermo Fisher Scientific) according to the manufacturer's multi-well plate protocol. For quality control, all samples were run on an SDS–PAGE gel and only samples with a clear band at 50 kDa were taken for downstream kinetic analyses.

### High-throughput determination of rubisco carboxylation rates

Purified rubisco samples were diluted to 800 nM and incubated with 4% CO$_2$ and 0.2% O$_2$ for rubisco activation (15 min; 4°C; plate shaker at 250 rpm). Rubisco catalysis requires activation by carbamylation of an active-site lysine residue (Pierce *et al*, 1980), which is achieved by high concentrations of CO$_2$. Following activation, 10 μl of the activated rubisco sample was added to six aliquots of 80 μl of assay mix (a detailed list of all assay components and their sources are provided in Appendix Table S1), each containing a different concentration of CABP, corresponding to 0, 0, 10, 20, 30, and 90 nM. Importantly, the assay mix was preincubated with 4% CO$_2$ and 0.2% O$_2$ to ensure constant gas composition throughout the procedure. Samples were incubated with CABP for 15 min to allow full inhibition (4% CO$_2$ and 0.2% O$_2$ at 30°C). Of note, that 0 nM conditions were performed in duplicate to more-accurately measure rubisco activity without inhibition. We chose 90 nM CABP as the highest concentration because it is sufficient for inhibiting all rubisco active sites (rubisco is diluted to a final concentration of 50 nM; see also Fig 2). Below, we describe the procedure used for CABP and RuBP synthesis and purification. To determine the carboxylation rate, NADH oxidation was monitored using a plate reader (Infinite® 200 PRO; TECAN) in UV-star plates (Greiner). The plate reader was connected to a gas control module (TECAN pro200) to establish atmospheric conditions of 4% CO$_2$ and 0.2% O$_2$; the temperature was set to 30°C to allow stable control of gas composition (according to the manufacturer's guidelines). To initiate rubisco carboxylation activity, 10 μl RuBP was added to each sample (final concentration of 1 mM and a total volume of 100 μl; the final concentration of the rubisco variants was 50 nM). The plate was then repeatedly shaken and measured for 340 nm (A$_{340}$) absorbance for 15 min every 2 s. RuBP was added to the plate using a multichannel repeater, which enabled us to initiate the reaction for all candidates at the same time and with minimal time gaps between replicates. NADH oxidation rates were calculated by fitting a linear regression model (custom python script; see below) to A$_{340}$ as a function of time and using an empirically determined effective extinction coefficient for NADH absorbance in our system

(Appendix). Because we used a multi-well plate (as opposed to a standard cuvette), our path length was different than 1 cm and, therefore, the reported extinction coefficient of NADH had to be adapted. To convert from NADH to reactions of rubisco per second, a stoichiometric ratio of 1:1 between NADH and 3PG was assumed, which translated to a stoichiometric ratio of 2:1 between NADH and carboxylation reactions (one carboxylation produces two 3PG molecules). For active-site quantification, we fit a linear regression model (custom python script; see below) to the measured reaction rates as a function of their associated CABP concentrations. Active-site concentration was indicated by the intercept with the *x*-axis of the fit line (Fig 2). To obtain the rate per active site, the activity with no CABP (*y*-intercept) was divided by the concentration of active sites (*x*-intercept).

### Radiometric kinetic analysis and CO$_2$:O$_2$ specificity assays

$^{14}$CO$_2$ fixation assays (0.5 ml reaction volume) were carried out at 25°C in 7.7 ml septum-capped glass scintillation vials (Wilson *et al*, 2018). Form-II rubiscos were purified similarly to the high-throughput expression protocol. For *S. elongatus*, we employed the plasmid and purification method from Mueller-Cajar and Whitney (2008). The assay buffer (100 mM EPPES-NaOH, pH 8.0, 20 mM MgCl$_2$, 1 mM EDTA) and all other components were equilibrated with N$_2$ gas. Reactions contained 10 μg/ml carbonic anhydrase, 1 mM RuBP, and $^{14}$CO$_2$ concentrations varying from 4 to 55 mM NaH$^{14}$CO$_3$ (corresponding to 50–700 μM $^{14}$CO$_2$). Purified rubisco (≈ 5 μM active sites) was first activated in an assay buffer containing 30 mM NaHCO$_3$. Twenty microliter aliquots of activated rubisco were used to initiate the assay, which was stopped after 2 min using 200 μl 50% (v/v) formic acid. The specific activity of $^{14}$CO$_2$ was measured by performing 30-minute assays containing 5.2 nanomoles of RuBP using the highest $^{14}$CO$_2$ concentration and ranged from 150 to 300 CPM/nmol RuBP. Reactions were dried using a heat block, resuspended in 750 μl water and 1 ml Ultima Gold XR scintillant before quantification using a scintillation counter. Rubisco active sites were quantified for each data set by performing [$^{14}$C]-2-CABP binding assays on 20 μl activated rubisco (Kubien *et al*, 2011). The $^{14}$C-CABP-bound rubisco was separated from the free ligand by size-exclusion chromatography and quantified by scintillation counting.

CO$_2$:O$_2$ specificity (S$_{C/O}$) assays were carried out at 25°C as detailed by Kane *et al* (1994) and Wilson and Hayer-Hartl (2018). Purified rubiscos were assayed in 20-ml septum-capped glass scintillation vials containing 1 ml 30 mM triethanolamine, pH 8.3, 15 mM Mg acetate, and 10 μg/ml carbonic anhydrase. The reactions were equilibrated in a defined gas mixture (995,009 ppm O$_2$; 4,991 ppm CO$_2$, Air Liquide Singapore) prior to the addition of [1-$^3$H]-RuBP. The reaction products were dephosphorylated using alkaline phosphatase (10 U/reaction), and radiolabeled glycerate and glycolate were separated on an HPX-87H column (Bio-Rad). The corresponding peaks were quantified using liquid scintillation counting, and S$_{C/O}$ was calculated as described by Kane *et al* (1994).

### RuBP and CABP synthesis, purification, and concentration determination

Since CABP is not commercially available and commercial RuBP is known to be contaminated with rubisco inhibitors (Kane *et al*,

1998; Andralojc *et al*, 2012), RuBP and CABP were enzymatically synthesized and purified in-house (Pierce *et al*, 1980). RuBP was synthesized enzymatically from D-ribose 5-phosphate and purified using anion-exchange chromatography as described in Kane *et al* (1998). RuBP was quantified using the spectrophotometric rubisco assay by measuring the reduction in absorbance at 340 nm. Two batches of CABP were synthesized from RuBP and both cold and [14]C-labeled KCN (Pierce *et al*, 1980). This reaction yields a mixture of CABP and the loose binding stereoisomer CRBP, which does not affect the analysis (Kubien *et al*, 2011). The specific activity of [14]C-CABP was equivalent to that of the [14]C-KCN batch used. The concentration of the cold CABP was determined using spectrophotometric rubisco assays, where a known concentration (determined by [14]C-CABP binding assay) of activated rubisco was inhibited by varying amounts of CABP. [1-[3]H]-RuBP was synthesized enzymatically from D-[2-[3]H]-glucose (PerkinElmer, Singapore) as described by Kane *et al* (1994).

### Computational pipeline availability

All the code used for generating our list of rubisco variants supporting carbon fixation, as well as the pipeline for analyzing the results of the enzymatic coupled assay, is open source and can be found in the following link: https://github.com/milo-lab/rubiscolympics.git.

**Expanded View** for this article is available online.

### Acknowledgements

We thank Dan Tawfik, Rui Alvez, Dina Listov, Natalie Page, Elad Noor, and Ed Bayer for productive feedback on this manuscript. Lynette Liew purified RuBP. We further thank the anonymous reviewers for comments that have made this manuscript clearer and better. This research was supported by the European Research Council (Project NOVCARBFIX 646827), the Israel Science Foundation (Grant 740/16), the Singapore National Research Foundation (NRF2017-NRF-ISF002-2667), the Beck-Canadian Center for Alternative Energy Research, Dana and Yossie Hollander, the Ullmann Family Foundation, the Helmsley Charitable Trust, the Larson Charitable Foundation, the Wolfson Family Charitable Trust, Charles Rothschild, and Selmo Nussenbaum. D.D is an EMBO long-term fellow (ALTF 1146-2018), an HFSP fellow (LT000232/2019-L) and a Rothschild fellow. R.M. is the Charles and Louise Gartner Professional Chair. Y.M.B.-O is an Azrieli Fellow.

### Author contributions

DD, MS, YMB-O, NP, AF, DGW, NA, OM-C, and RM were involved in the conceptualization of this work. Data curation was performed by DD, MS, ZG, YMB-O, BdP, LS, IS, OM-C, and RM. Formal analysis was done by DD, MS, ZG, YMB-O, NP, AO, JJ, DGW, DH, YP, SA, IS, and OM-C. Investigation was done by DD, MS, ZG, YMB-O, NP, DWG, IS, OM-C, and RM. DD, MS, YMB-O, NP, AF, DGW, OM-C, and RM developed the methodology of this work. Code was written by DD, YMB-O, AO, and IS. Visualization was performed by DD, YMB-O, OM-C, and RM. Validation of data was performed by DD, MS, and RM. Writing was done by DD, AF, OM-C, and RM. Review and editing were done by DD, YMB-O, AF, OM-C, and RM. RM and OM-C supervised the work.

### Conflict of interest

The authors declare that they have no conflict of interest.

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
