## [Review Process File · The EMBO Journal]

Highly active rubiscos discovered by systematic interrogation of natural sequence diversity

Ron Milo, Dan Davidi, Melina Shamshoon, Zhijun Guo, Yinon Bar-On, Noam Prywes, Aia Oz, Jagoda Jablonska, Avi Flamholz, David Wernick, Niv Antonovsky, Benoit de Pins, Lior Shachar, Dina Hochhauser, Yoav Peleg, Shira Albeck, Itai Sharon, and Oliver Mueller-Cajar
DOI: [embj.2019104081](https://doi.org/10.1042/emboj.2019104081)

Corresponding author(s): Ron Milo (Ron.Milo@weizmann.ac.il)

Review Timeline:

Submission Date:	23rd Nov 19
Editorial Decision:	20th Dec 19
Appeal Received:	27th Jan 20
Editorial Decision:	6th Feb 20
Revision Received:	27th Feb 20
Editorial Decision:	15th Apr 20
Revision Received:	30th Apr 20
Accepted:	7th May 20

Editor: Ieva Gailite

Transaction Report:

Thank you for submitting your manuscript for consideration by The EMBO Journal. We have now received two referee reports on your manuscript, which are included below for your information. While the third reviewer was not able to submit their comments yet, since the two reports are in fair agreement, I am taking the decision based on the comments at hand to avoid further delays over the holiday period. Based on the reviewer comments, we unfortunately had to conclude that the study is not a sufficiently strong candidate for publication in The EMBO Journal.

As you can see, while both reviewers find the topic of interest, they also indicate substantial limitations in the analysis of the enzymatic properties of the described novel Rubiscos and find that the described enzymes are likely to have limited applicability for improvement of plant productivity. Both reviewers also indicate in the overall manuscript evaluation that the advance provided is of only moderate novelty and general interest. Given these opinions from good experts in the research field, I am afraid that we cannot offer further proceedings towards publication in The EMBO Journal.

That being said, I realise that the study will be of interest to other researchers in the field, and I would like to propose a transfer of your manuscript and referee reports to our sister journal Life Science Alliance. I have already discussed your work with Andrea Leibfried, executive editor of Life Science Alliance, and Andrea would like to offer publication of your manuscript after a minor revision. She would expect a point-by-point response and accordingly changes in the manuscript to appropriately tone down the statements and provide a more balanced discussion of the results. Furthermore, she would like to request further analysis of the Rubisco kinetics in presence of higher levels of O₂, if possible. Andrea would be happy to discuss the process and the scope of the revision at any point - you can contact her at a.leibfried@life-science-alliance.org.

Thank you in any case for the opportunity to consider this manuscript. I am sorry that I could not communicate more positive news, but I nevertheless hope that you will find our reviewers' comments helpful and that you will find the transfer option of interest. Please also note that the transfer link does not have an expiry date, and you can return to this option at a later time point.

Referee #1:

This manuscript describes a creative and relatively unique initiative to screen a broad assortment of Form II and Form II/III Rubisco isoforms to ascertain the level of natural diversity in their rates of CO₂-fixation (kcat). The candidate enzymes tested were selected following phylogenetic analyses of Rubisco large subunit sequences derived from mining metagenomic data. 143 Rubisco

sequences spanning differing phylogenetic regions were chosen and the k_{cat} for 78 successfully obtained to reveal a span in diversity greater than currently perceived. Higher resolution k_{cat} and K_mCO_2 measurements on 9 select enzyme variants (including appropriate *R. rubrum* and *S. elongatus* controls) revealed 5 Form II isoforms with carboxylation properties superior to Form I *S. elongatus* Rubisco, the fastest isoform known to date.

The manuscript is very easy to read and the modified plate based assay will be of significant interest to those wishing to measure k_{cat} via a higher throughput process to the standard $14CO_2$ -fixation approach. Below are a number of comments for the authors to consider. As a general comment, the authors make a number of grandiose assertions as to the impact of the work in the context of improving plant productivity. As detailed below, there are flaws in some of these statements and if unchecked would add further fuel to the growing misconceptions in the literature that these enzymes have potential use in plant productivity enhancement. Their usefulness in such an application would require a highly efficient CCM (which they briefly touch on in the discussion) and that their sensitivity to oxygenation inhibition was not too detrimental (which was not assessed here).

Here are some relatively minor comments the authors should consider/address:

ABSTRACT

1. What defines a catalytically superior Rubisco is highly context dependent. In plants and algae a Rubisco with a high CO_2/O_2 specificity would broadly be described as superior, somewhat independent of k_{cat} . So for the authors to say they are searching for "catalytically superior variants" is misplaced and at odds with what they actually measured for which is actually for Rubisco with higher (substrate saturated) rates of CO_2 -fixation. (Notably in nature the CO_2 fixation rates reach would unlikely be anywhere close to the saturated rates without a cyanobacteria equivalent CCM given the high K_mCO_2 's required).
2. While 143 Rubisco genes were synthesized, only 78 were actually purified and biochemically tested.
3. Not sure of the reasoning as to why these enzymes would be "especially promising for biotechnology" or for "feeding a growing population" (a rather grandiose, unsubstantiated statement)?

INTRODUCTION

4. No mention of the Rubisco oxygenation reaction is mentioned. The focus is primarily on Rubisco being a "slow" as its only kinetic constraint. For the novice reader it would be unclear as to the impact O_2 inhibition influences catalysis - one of the main reasons it is such an intensively studied enzyme. and a major constraint on Rubisco catalytic efficiency (its not just a "slow" enzyme).
5. P2 - replace "made relatively little progress" with "made limited progress". It might be worth mentioning that (real) successes in improving the net carboxylation capacity of Rubisco have only been achieved for Form I and III isoforms (the latter by Wilson et al., 2016, Scientific Report) and only by directed evolution. (See also recent paper by Yu and Whitney, 2019, Int J Mol Sci 20: 5019)
6. The authors should again be cautious in their terminology. In the last paragraph on P2, to be more accurate the text should state "in search of catalytic outliers in terms of CO_2 fixation speed and CO_2 affinity". Similarly, what defines "superior CO_2 fixation kinetics" is arbitrary. For example, according to the data in table 2 the carboxylation efficiency of the enzymes analyzed in this study are all 3-10 fold lower than plant Form I Rubisco and so one could say none are "superior". Certainly one would anticipate the CO_2/O_2 specificity of eukaryote sources of Form I Rubiscos would similarly be "superior" to those measured in this study.

RESULTS

7. Does the assertion on P3 that additional metagenomic unveiling of Rubisco sequence diversity will diminish the dominance of Form I Rubisco sequence diversity take into account the sequence diversity (and multi-gene copies) of RbcS?
8. P4; Suggest rewording to "folding and assembly of plant form-I rubisco" to align with the

reference cited.

9. P4; "4 failed to express in *E. coli*". Please clarify if referring to there being no evidence of functional enzyme biogenesis (solubility), or no detection of soluble LSu made (which would suggest a cloning error).

10. P4; replace "3-phosphoglycerate will be formed" with "2-phosphoglycolate will be formed".

11. Not sure why Table 2 is in the SI, it is more suited to the main text and could easily replace Fig 2 that could be moved to the SI (but I would note it is a very clear and nicely presented figure).

12. Additional columns should be included in Figure 2 to include the following additional information.

(a) The RbcL subunit stoichiometry of each enzyme. (b) the carboxylation efficiency values (i.e. k_{cat}/K_M) at 25°C. (c) the corresponding k_{cat} derived by the spectrophotometric assay (for comparative assessment of accuracy, recognising they were done at differing temps). (d) inclusion of comparative examples of literature values for plant Form I Rubisco (so that readers can clearly ascertain that while the form II and cyanobacteria enzymes may be faster, their low CO₂-affinities compromise their carboxylation efficiencies).

DISCUSSION

13. P8; arguably only tested 78, not 143, Rubisco variants were tested.

14. It is stated that one basal clade of organisms tended to have higher k_{cat} s, however it is impossible to ascribe in Figure 2 any correlation with the enzymes analysed in table 2. Also lacking is any description of the comparative amino acid similarity among the Form II enzymes characterized in Table 2? Is there sufficient similarity for the authors to start speculating about specific RbcL amino acids that influence k_{cat} ?

15. To make the assertion that expression of Gallionella Rubisco would not be problematic in organisms with a CCM is making the assumption their Sc/o is not too crippled. Based on prior publications from this group one might suspect the Sc/o of the high k_{cat} enzymes might be particularly impaired. As modelled by Sharwood (2017, *New Phytologist*), such enzymes would still perform more poorly in a C4 plant than that native plant Rubisco.

16. Replace "them to target the fast" to "them integrate the fast"

17. The last paragraph needs to acknowledge the limitations of the study. This study only focused on one facet of Rubisco kinetics (CO₂-fixation speed and affinity) - it did not examine the properties of its other two substrates (RuBP and O₂), both of which are needed to truly determine the relative "superiority" of a Rubisco in differing physiological contexts - in particular within C3 and C4 plants in which appears the underpinning objective based on the last sentence of the abstract.

Referee #2:

Overall, this is a very solid study that focuses on screening the kinetic diversity of RubisCO's carboxylation reaction across the phylogenetic tree. The study extends on earlier efforts in the field, in particular by the Tabita, Perner and Kerfeld labs. Compared to the reports from the Tabita (several publications over the years) and Perner labs (*ISME Journal* 2019), this study comprises a bigger data set. Its exclusive focus on existing RubisCOs makes it different from the Kerfeld study (*Nat. Comm.* 2017) that additionally took evolutionary considerations into account (through assessing the kinetic diversity of existing and extent RubisCOs by ancestral reconstruction from phylogenetic trees). In summary, the manuscript by Davidi et al. complements very nicely these other publications.

The experiments are thoroughly designed and the data is of high quality. The study has its merits for (more) systematically extending the set of kinetic data for RubisCO. However, the study does not find any correlation patterns between the kinetic parameters of individual RubisCO homologs

and phylogenetic or environmental signatures, which would have been very exciting and a major advancement for the field (note that the Kerfeld study did identify some trends).

The authors claim that they were able to identify a RubisCO homologue that is two times faster than any other RubisCO homolog reported before. First of all, the Tabita lab reported already 2006 on a type III RubisCO with 23 s⁻¹ (although at elevated temperature). Second of all, such findings are not uncommon for (medium sized) phylogenetic screens (see related work from enzyme screening labs). Also note that the reported CO₂-fixation activity is still almost five-fold lower than that of other CO₂-fixing enzymes (especially PEP carboxylase and crotonyl-CoA carboxylase/reductase). Overall, these findings are not unexpected and do not essentially change our understanding of RubisCO and its catalytic limitations compared to other CO₂ fixing enzymes. There are also some experimental doubts on the reported "fastest RubisCO" (see below, detailed comments).

Most importantly, the study fails in respect to a crucial point. Note that one important kinetic parameter in RubisCO is carboxylation activity. An equally important parameter, however, is the specificity of RubisCO to discriminate between CO₂ and O₂. It has been reported several times that RubisCO shows an inverse correlation ('trade-off') between carboxylation activity and specificity. The faster a RubisCO homolog, the lower is typically its specific reaction with CO₂. This study does not report on a single RubisCO that apparently breaks this inverse correlation, which would have been challenging the longstanding dogma in the field and be of immediate relevance for any efforts to improve photosynthetic yield in plants. In other words, the two-fold faster RubisCO is not really useful, if it is not also more specific for CO₂. Finding a faster and more specific RubisCO is a holy grail, but this study did not provide such a long-sought after enzyme.

In summary, this study is a very valuable contribution to the field of RubisCO biochemistry and a step forward, but a rather an incremental one.

More detailed comments on the manuscript:

Abstract: Please remove the statement that the identified enzymes will contribute to the challenge of feeding a growing world population. This is too far-fetched and the enzymes identified show extremely high K_m values for CO₂, which would be a huge challenge for any efforts of implementing them into plants.

Introduction: "It has been suggested that the catalytic properties of this enzyme may already be optimized and thus cannot be straight forwardly improved." This is a misleading statement, because this argument is only valid if one considers oxygenation versus carboxylation, which is in fact not considered in this study. When only looking at carboxylation activity in anaerobic or carbon concentrating mechanisms, there had been RubisCOs reported with 14 s⁻¹ (*Synechococcus*) and 23 s⁻¹ (*Archaeoglobus*), which clearly indicates that there is less constraints, if the oxygen side reaction can be neglected.

Figure 5: The data on the *Galionella* enzyme seems to be very sensitive to the fitting, as the enzyme was not measured under fully saturating conditions. It rather looks like the enzyme would level off at 16 s⁻¹, which would be much closer to the 14s⁻¹ reported for the *Synechococcus* enzyme, but this remains unclear because the data at higher concentrations was not collected (and not experimentally compared to the *Synechococcus* enzyme as reference).

Referee #1:

This manuscript describes a creative and relatively unique initiative to screen a broad assortment of Form II and Form II/III Rubisco isoforms to ascertain the level of natural diversity in their rates of CO₂-fixation (k_{cat}). The candidate enzymes tested were selected following phylogenetic analyses of Rubisco large subunit sequences derived from mining metagenomic data. 143 Rubisco sequences spanning differing phylogenetic regions were chosen and the k_{cat} for 78 successfully obtained to reveal a span in diversity greater than currently perceived. Higher-resolution k_{cat} and K_mCO₂ measurements on 9 select enzyme variants (including appropriate *R. rubrum* and *S. elongatus* controls) revealed 5 Form II isoforms with carboxylation properties superior to Form I *S. elongatus* Rubisco, the fastest isoform known to date. The manuscript is very easy to read and the modified plate-based assay will be of significant interest to those wishing to measure k_{cat} via a higher throughput process to the standard ¹⁴C-fixation approach.

Below are a number of comments for the authors to consider. As a general comment, the authors make a number of grandiose assertions as to the impact of the work in the context of improving plant productivity. As detailed below, there are flaws in some of these statements and if unchecked would add further fuel to the growing misconceptions in the literature that these enzymes have potential use in plant productivity enhancement. Their usefulness in such an application would require a highly efficient CCM (which they briefly touch on in the discussion) and that their sensitivity to oxygenation inhibition was not too detrimental (which was not assessed here).

The paper was indeed somewhat over-hyped in its assertions as a result of being sent in the previous round to *Nature*. We now made changes in the manuscript to appropriately tone down the statements and provide a more balanced discussion of the results. We feel this was a very correct criticism that make us much happier with the updated paper.

Here are some relatively minor comments the authors should consider/address:

ABSTRACT

1. What defines a catalytically superior Rubisco is highly context-dependent. In plants and algae, a Rubisco with a high CO₂/O₂ specificity would broadly be described as superior, somewhat independent of k_{cat}. So for the authors to say they are searching for "catalytically superior variants" is misplaced and at odds with what they actually measured for which is actually for Rubisco with higher (substrate saturated) rates of CO₂-fixation. (Notably, in nature the CO₂ fixation rates reach would unlikely be anywhere close to the saturated rates without a cyanobacteria equivalent CCM given the high K_mCO₂'s required).

The term "catalytically superior variants" was replaced with "fast carboxylating variants".

2. While 143 Rubisco genes were synthesized, only 78 were actually purified and biochemically tested.

We now have a collection of 105 purified and active enzymes. We now present the results of all of them and explain that some fraction (much smaller than before) were insoluble.

3. Not sure of the reasoning as to why these enzymes would be "especially promising for biotechnology" or for "feeding a growing population" (a rather grandiose, unsubstantiated statement)?

We apologize for the originally grandiose text. We removed this statement.

INTRODUCTION

4. No mention of the Rubisco oxygenation reaction is mentioned. The focus is primarily on Rubisco being a "slow" as its only kinetic constraint. For the novice reader, it would be unclear as to the impact O₂ inhibition influences catalysis - one of the main reasons it is such an intensively studied enzyme. and a major constraint on Rubisco catalytic efficiency (its not just a "slow" enzyme).

Oxygenation is indeed an important aspect of rubisco's kinetics. Since the initial submission of the manuscript, we have measured the specificity factor ($S_{C/O}$) for our fastest 7 candidates and included it in the results section. Further and given the referee's comment, we now also elaborate on the specificity-rate tradeoff of rubisco.

5. P2 - replace "made relatively little progress" with "made limited progress". It might be worth mentioning that (real) successes in improving the net carboxylation capacity of Rubisco have only been achieved for Form I and III isoforms (the latter by Wilson et al., 2016, Scientific Report) and only by directed evolution. (See also the recent paper by Yu and Whitney, 2019, Int J Mol Sci 20: 5019)

The wording was updated according to the referee's comment. We also appreciate the pointers provided by the referee and have now included them in the manuscript.

6. The authors should again be cautious in their terminology. In the last paragraph on P2, to be more accurate the text should state "in search of catalytic outliers in terms of CO₂ fixation speed and CO₂ affinity". Similarly, what defines "superior CO₂ fixation kinetics" is arbitrary. For example, according to the data in table 2 the carboxylation efficiency of the enzymes analyzed in this study are all 3-10 fold lower than plant Form I Rubisco and so one could say none are "superior". Certainly one would anticipate the CO₂/O₂ specificity of eukaryotic sources of Form I Rubiscos would similarly be "superior" to those measured in this study.

In retrospect we fully agree with the referee and updated the wording as well as added: "... in terms of CO₂ fixation speed (k_{cat} for carboxylation)." to the text.

RESULTS

7. Does the assertion on P3 that additional metagenomic unveiling of Rubisco sequence diversity will diminish the dominance of Form I Rubisco sequence diversity take into account the sequence diversity (and multi-gene copies) of RbcS?

The analysis regarding the sequence diversity of rubisco was based only on the large subunit. We agree that not accounting for the diversity of the small subunit can skew the degree of sequence diversity in Form I rubiscos compared to all other forms that do not have a small subunit. We now make it clear in the text that the analysis was based on the large subunit and changed the wording of the above sentence to:

"Notably, while 97% of the sequenced rubiscos are form-I, they account for only ≈50% of the sequence diversity of rubisco's large subunit. It is likely that future metagenomic efforts will uncover novel rubisco sequences that shall further expand the contribution of form-II, III and II/III rubiscos to the global sequence diversity of this enzyme."

8. P4; Suggest rewording to "folding and assembly of plant form-I rubisco" to align with the reference cited.

Done.

9. P4; "4 failed to express in *E. coli*". Please clarify if referring to there being no evidence of functional enzyme biogenesis (solubility), or no detection of soluble LSu made (which would suggest a cloning error).

All plasmids cloned for this study were sequence verified (by Sanger and/or Next-Generation sequencing). The term "failed to express" was used instead of the appropriate "insoluble". Accordingly, we changed the wording to:

"Out of 18 tested variants, 4 were insoluble..."

10. P4; replace "3-phosphoglycerate will be formed" with "2-phosphoglycolate will be formed".

The enzymatic coupled assay used here couples 3-phosphoglycerate production to oxidation of NADH. With that respect, the oxygenation reaction is problematic because 3-phosphoglycerate is also formed (alongside 2-phosphoglycolate). This is why if "3-phosphoglycerate will be formed via a promiscuous oxygenation reaction", the coupled assay would be biased. To make it more clear, we now write:

"A potential caveat to this assay is that if O₂ levels are sufficiently high, 3-phosphoglycerate will be formed via a promiscuous oxygenation reaction (alongside 2-phosphoglycolate), resulting in a deviation between the rates of NADH oxidation and carboxylation."

11. Not sure why Table 2 is in the SI, it is more suited to the main text and could easily replace Fig 2 that could be moved to the SI (but I would note it is a very clear and nicely presented figure).

This was the result of tight space constraint in the previous submission. We agree with the referee and have now moved Table S2 to the main text. As elaborated in the point below, we also included further columns to the table.

12. Additional columns should be included in Table 2 to include the following additional information. (a) The RbcL subunit stoichiometry of each enzyme. (b) the carboxylation efficiency values (i.e. kcat/KM) at 25oC. (c) the corresponding kcat derived by the spectrophotometric assay (for comparative assessment of accuracy, recognising they were done at different temps). (d) inclusion of comparative examples of literature values for plant Form I Rubisco (so that readers can clearly ascertain that while the form II and cyanobacteria enzymes may be faster, their low CO₂-affinities compromise their carboxylation efficiencies).

Following the referee's comment, (a) and (b) and (d) were added to the Table which is now in the main text. We further included the specificity-factor ($S_{c/o}$) of each variant, as we were able to determine them since the initial submission of the manuscript. The corresponding specific activities derived by the spectrophotometric assay is performed under a different temperature and level of CO₂ saturation and thus can be confusing when out of context - we give a

comparison with the description of those issues in the SI. Importantly, we added the following paragraph to the main text:

“The specificity factor of the set of 7 fast rubiscos was also measured to reflect the relative capacities of these enzymes to catalyze carboxylation over oxygenation of ribulose 1,5-bisphosphate. Compared to *S. elongatus*, which has a specificity factor of 42.7 ± 2.8 , L₂ rubisco from *Gallionella* sp. is fourfold less specific (10.0 ± 0.1). In general, the specificity of all our bacterial rubiscos was significantly lower than *S. elongatus*, which is already more than twofold less specific than the median plant rubisco (plant variants exhibit a specificity factor range of ≈ 80 -120 (Flamholz et al, 2018)). These measurements thus indicate that in order to support considerable carboxylation flux, our novel rubiscos would have to operate in the context of a CCM or in otherwise high CO₂ environments.”

DISCUSSION

13. P8; arguably only tested 78, not 143, Rubisco variants were tested.

We now have activity values for 105 variants out of 143 in the initial library. The sentence was changed to:

“Here we tested >100 rubisco variants...”

14. It is stated that one basal clade of organisms tended to have higher kcats, however, it is impossible to ascribe in Figure 2 any correlation with the enzymes analysed in table 2.

Following the comment, we now annotate the bars in Figure 2 that correspond to the 7 variants that were further analyzed.

Also lacking is any description of the comparative amino acid similarity among the Form II enzymes characterized in Table 2? Is there sufficient similarity for the authors to start speculating about specific RbcL amino acids that influence kcat?

The clustering algorithm used to choose representative sequences in this study found variants that are less than 90% identical to one another at the amino acid level (which results in covering the sequence space at a resolution of 90% sequence-identity). This means that despite being in close proximity on the dendrogram in Figure 2, each variant is at least 40 amino-acids different than any other variant in the set (the length of rubisco is ≥ 400 a.a).

Unfortunately, despite being an interesting question raised by the referee, the relatively large sequence distance alongside the small dynamic range in kinetic properties was insufficient to speculate/conclude anything compelling. That being said, we fully agree with the referee that looking at the relationship between sequence-similarity and kinetic-similarity is valuable for studying rubisco as well as other enzymes (and, in fact, we wrote about that in a recent review (Davidi et al. 2018)). In a followup work we are now looking into other members of the clusters of those fast variants. Hopefully, analysis at higher resolution would allow us to identify positions that are important for catalysis, that could be specifically targeted in future studies.

15. To make the assertion that expression of Gallionella Rubisco would not be problematic in organisms with a CCM is making the assumption their $S_{C/O}$ is not too crippled. Based on prior publications from this group one might suspect the $S_{C/O}$ of the high k_{cat} enzymes might be particularly impaired. As modelled by Sharwood (2017, New Phytologist), such enzymes would still perform more poorly in a C4 plant than that native plant Rubisco.

Having measured also $S_{C/O}$, we now know that these present relatively low affinity to CO_2 and $S_{C/O}$ values. Indeed, in the context of C_4 plants, that will not be sufficient to elevate fixation rates. That being said, tradeoffs in rubisco kinetics seem to be relaxed for variants that act in organisms with CCMs (Iñiguez et al. 2020). It is possible that the fast variants discovered here would be a promising starting point for directed evolution in CCM containing model systems.

16. Replace "them to target the fast" to "them integrate the fast"

Done.

17. The last paragraph needs to acknowledge the limitations of the study. This study only focused on one facet of Rubisco kinetics (CO₂-fixation speed and affinity) - it did not examine the properties of its other two substrates (RuBP and O₂), both of which are needed to truly determining the relative "superiority" of a Rubisco in differing physiological contexts - in particular within C₃ and C₄ plants in which appears the underpinning objective based on the last sentence of the abstract.

As indicated by the referee, the fast variants discovered here exhibit limitations. Specifically we report relatively low affinity to CO_2 and $S_{C/O}$ values. Indeed, in the context of C_4 plants, that will not be sufficient to elevate fixation rates. On the other hand, and as pointed out in our response to point (15), tradeoffs in rubisco kinetics seem to be relaxed for variants that act in organisms with CCMs (Iñiguez et al. 2020). Our variants ,therefore, may prove to be a promising starting point for directed evolution in CCM containing model systems. Upon the removal of several overhyped statements in the manuscript (including in the abstract), we believe that along with major edits (as can be seen in the main text), the last paragraph of the discussion now accurately reflects the findings in the presented study.

Referee #2:

Overall, this is a very solid study that focuses on screening the kinetic diversity of RubisCO's carboxylation reaction across the phylogenetic tree. The study extends on earlier efforts in the field, in particular by the Tabita, Perner and Kerfeld labs. Compared to the reports from the Tabita (several publications over the years) and Perner labs (ISME Journal 2019), this study comprises a bigger data set. Its exclusive focus on existing RubisCOs makes it different from the Kerfeld study (Nat. Comm. 2017) that additionally took evolutionary considerations into account (through assessing the kinetic diversity of existing and extent RubisCOs by ancestral reconstruction from phylogenetic trees). In summary, the manuscript by Davidi et al. complements very nicely these other publications.

The experiments are thoroughly designed and the data is of high quality. The study has its merits for (more) systematically extending the set of kinetic data for RubisCO. However, the study does not find any correlation patterns between the kinetic parameters of individual RubisCO homologs and phylogenetic or environmental signatures, which would have been very exciting and a major advancement for the field (note that the Kerfeld study did identify some trends).

The authors claim that they were able to identify a RubisCO homologue that is two times faster than any other RubisCO homolog reported before. First of all, the Tabita lab reported already 2006 on a type III RubisCO with 23 s⁻¹ (although at elevated temperature). Second of all, such findings are not uncommon for (medium sized) phylogenetic screens (see related work from enzyme screening labs).

The Tabita lab indeed found a form-III rubisco that catalyze 23 s⁻¹ from the hyperthermophiles archaea *A. fulgidus*. We now cite this paper explicitly in the text. However, this value was measured at 83°C. At 25°C, this enzyme was marginally active in our hands, and indeed, if we correct the reported value using $Q_{10}=2.2$ (Cen & Sage, 2005), the k_{cat} at 25°C would be ≈ 0.75 s⁻¹.

Also note that the reported CO₂-fixation activity is still almost five-fold lower than that of other CO₂-fixing enzymes (especially PEP carboxylase and crotonyl-CoA carboxylase/reductase). Overall, these findings are not unexpected and do not essentially change our understanding of RubisCO and its catalytic limitations compared to other CO₂ fixing enzymes. There are also some experimental doubts on the reported "fastest RubisCO" (see below, detailed comments).

We fully agree with the referee that other carboxylases have significantly faster carboxylation rates compared to rubisco. Part of it may be due to the fact that rubisco uses CO₂ as a substrate instead of bicarbonate. In the past, we and others suggested using PEP carboxylase as part of a synthetic CO₂ fixation pathway. Such efforts are being pursued by several labs including Tobias Erb's lab at the Max Planck institute. Unfortunately though, making PEP carboxylase work *in-vivo* in an autocatalytic flux-mode for CO₂ fixation is highly challenging. Form II rubiscos, on the other hand, have been shown to support CO₂ fixation flux when heterologously expressed in autotrophic organisms as well as in synthetic systems in heterotrophic microorganisms (e.g. Gleizer et al., *Cell*, 2019). It is indeed an invaluable effort of the community to try other carboxylases for CO₂ fixation *in-vivo*, yet we believe that synthetic rubisco-based CO₂ fixation in living organisms is still a highly valuable avenue to try at this point in time.

Most importantly, the study fails in respect to a crucial point. Note that one important kinetic parameter in RubisCO is carboxylation activity. An equally important parameter, however, is the specificity of RubisCO to discriminate between CO₂ and O₂. It has been reported several times that RubisCO shows an inverse correlation ('trade-off') between carboxylation activity and specificity. The faster a RubisCO homolog, the lower is typically its specific reaction with CO₂. This study does not report on a single RubisCO that apparently breaks this inverse correlation, which would have been challenging the longstanding dogma in the field and be of immediate relevance for any efforts to improve photosynthetic yield in plants. In other words, the two-fold faster RubisCO is not really useful, if it is not also more specific for CO₂. Finding a faster and more specific RubisCO is a holy grail, but this study did not provide such a long-sought-after enzyme.

Having now measured the specificity factor of our top 7 variants as we report in the updated text and table, we now know that their CO₂:O₂ discrimination capacity is low but still in the ballpark of

efficient carboxylation kinetics at physiologically relevant CO₂ concentrations in organisms with CCM.

As the referee pointed out, finding rubisco variants that break the tradeoff line would represent a holy grail and utmost exciting news. While we did not find such rubiscos in this study, we believe that the presented approach is a valuable path toward discovering such variants if they exist, especially in light of our discoveries of the highest kcat carboxylating rubiscos to date at 25° C. Specifically we view the bioinformatic pipeline as well as the plate-based carboxylation assay as useful contributions to the field as they allow to systematically explore the kinetic space of rubisco at unprecedented throughput.

In summary, this study is a very valuable contribution to the field of RubisCO biochemistry and a step forward, but rather an incremental one.

More detailed comments on the manuscript:

Abstract: Please remove the statement that the identified enzymes will contribute to the challenge of feeding a growing world population. This is too far-fetched and the enzymes identified show extremely high Km values for CO₂, which would be a huge challenge for any efforts of implementing them into plants.

This statement was removed from the text.

Introduction: "It has been suggested that the catalytic properties of this enzyme may already be optimized and thus cannot be straightforwardly improved." This is a misleading statement because this argument is only valid if one considers oxygenation versus carboxylation, which is in fact not considered in this study.

We have now added the following sentence to the introduction:

“Specifically, rubisco can use O₂ as a substrate instead of CO₂ to catalyze an oxygenation reaction. A tradeoff between greater CO₂:O₂ specificity (S_{C/O}) and a greater reaction rate was previously reported (Tcherkez et al, 2006; Savir et al, 2010). In this tradeoff, the enzyme might have reached what was discussed as a local fitness optimum.”

Further, as indicated above, since the initial submission of the manuscript, S_{C/O} was determined for the fastest 7 variants (reported in the updated results section).

When only looking at carboxylation activity in anaerobic or carbon concentrating mechanisms, there had been RubisCOs reported with 14 s⁻¹ (Synechococcus) and 23 s⁻¹ (Archaeoglobus), which clearly indicates that there is less constraints, if the oxygen side reaction can be neglected.

As was highlighted by the referee, organisms that live in micro-aerobic or anaerobic conditions can express rubisco homologs that are less kinetically constrained, as the oxygenation reaction can be neglected. Indeed, we state in the manuscript that among the fastest 7 variants, at least 5 live in low to no oxygen habitats.

Regarding other fast rubiscos that were previously reported: our understanding of the 23 s^{-1} value is detailed above. As for the 14 s^{-1} value reported for *S. elongatus*, it is markedly higher compared to 5 other papers that measured the rate of that same variant (either from *Synechococcus* PCC6301 or PCC7942, which encode for an identical enzyme). In fact, the second fastest value reported for *S. elongatus* is 11.8 s^{-1} and the median k_{cat} value for this enzyme is 11.6 s^{-1} .

In general, oftentimes we see remarkable variability across kinetic measurements between different papers (see more discussion e.g. in (Davidi et al. 2016)). To manage those variations we referred to the median k_{cat} value across all measurements, and at the same time, included rubisco from *S. elongatus* as a control in our isotopic labeling experiments.

Lastly, a k_{cat} value of 14 s^{-1} would still be about 40% lower than that of the fastest rubisco discovered in this study (22 s^{-1}).

Figure 5: The data on the *Galionella* enzyme seems to be very sensitive to the fitting, as the enzyme was not measured under fully saturating conditions. It rather looks like the enzyme would level off at 16 s^{-1} , which would be much closer to the 14 s^{-1} reported for the *Synechococcus* enzyme, but this remains unclear because the data at higher concentrations was not collected (and not experimentally compared to the *Synechococcus* enzyme as reference).

Going over dozens of kinetic reports of rubisco, the concentration of CO_2 used for kinetic measurements is up to 40 mM of NaHCO_3 , which would correspond to about $400 \mu\text{M}$ of CO_2 (at pH 8 and 25°C). This is also true for the reported studies for *S. elongatus*, which, on average report a median k_{cat} of 11.6 s^{-1} . Having a K_M value of about $200 \mu\text{M}$, determining the k_{cat} for *S. elongatus* at $400 \mu\text{M}$ of CO_2 is more sensitive to fitting than was done in our study ($500 \mu\text{M}$ of CO_2). Accordingly, at $500 \mu\text{M}$ CO_2 , the specific activity of *S. elongatus* would be about 8 s^{-1} compared to about 15 s^{-1} for the *Galionella* enzyme. It is important to note that higher concentrations of CO_2 are not easy to test, since going above $500 \mu\text{M}$ of CO_2 can result in substrate inhibition, as well as affect the pH of the solution, to which the carboxylation reaction of rubisco is sensitive. It is likely that rubisco studies have traditionally been using relatively low concentrations of CO_2 since plant rubiscos have been mostly assayed and their K_M values are on the order of $50 \mu\text{M}$. In general, in most if not all studies the k_{cat} is extrapolated from the data in a similar manner due to these constraints, and thus for all low affinity rubiscos the challenge of fitting similarly exists.

Thank you for contacting me with a preliminary revision plan for your manuscript. I have now received positive assessments of your revision proposal from both original reviewers, and they are interested in evaluating the revised manuscript. I would therefore like to invite you to submit a revised manuscript in which you address the comments of both reviewers as indicated in your preliminary point-by-point response. Please note that reviewer #1 also made additional comments that I have copied below, as they may be helpful for preparation of the revised version.

We generally allow three months as standard revision time. As a matter of policy, competing manuscripts published during this period will not negatively impact on our assessment of the conceptual advance presented by your study. However, please contact me as soon as possible upon publication of any related work to discuss how to proceed.

When preparing your letter of response to the referees' comments, please bear in mind that this will form part of the Review Process File, and will therefore be available online to the community. For more details on our Transparent Editorial Process, please visit our website: http://emboj.embopress.org/about#Transparent_Process

Please feel free to contact me if have any further questions regarding the revision. Thank you for the opportunity to consider your work for publication. I look forward to your revision.

Reviewer #1 additional comments:

There are still some statements in the revised version that may need to be addressed.

I would also point out Review 2 comments on Fig 5 are unjustified - they (and the authors here) don't seem to appreciate that assaying at saturating substrate levels is not critical when fitting a Michaelis-Menten fit. The critical component is ensuring accuracy around the K_m value as the initial slope of the fit actually represents the value for V_c/K_m (catalytic efficiency). So in essence, those who focus on getting V_c values using too-saturating [substrate]'s actually get lower V_c/K_m values which don't fit the experiment data.

One other comment is the authors comments on the $k_{cat} - S_{co}$ trade-off need to carefully consider this contention is largely outdated as being a constraint - and base around a very limited dataset. As the authors actually indicate, there is quite a bit of flexibility in this relationship among organisms with a CCM (e.g. C4-plants have 2-fold higher k_{cats} than C3-plant Rubisco but the same, or better, S_{co} e.g. Maize, sorghum etc) and certainly any such trade-off does not hold for improved

Rubiscos generated by directed evolution. Any evidence of the trade-off in aquatic organisms breaks down even more as their Rubisco have evolved in O₂ and CO₂ environments that vary by the hour depending where they find themselves in the water column, irrespective of whether or not they have a CCM.

Referee #1:

This manuscript describes a creative and relatively unique initiative to screen a broad assortment of Form II and Form II/III Rubisco isoforms to ascertain the level of natural diversity in their rates of CO₂-fixation (k_{cat}). The candidate enzymes tested were selected following phylogenetic analyses of Rubisco large subunit sequences derived from mining metagenomic data. 143 Rubisco sequences spanning differing phylogenetic regions were chosen and the k_{cat} for 78 successfully obtained to reveal a span in diversity greater than currently perceived. Higher-resolution k_{cat} and K_mCO₂ measurements on 9 select enzyme variants (including appropriate *R. rubrum* and *S. elongatus* controls) revealed 5 Form II isoforms with carboxylation properties superior to Form I *S. elongatus* Rubisco, the fastest isoform known to date. The manuscript is very easy to read and the modified plate-based assay will be of significant interest to those wishing to measure k_{cat} via a higher throughput process to the standard ¹⁴C-fixation approach.

Below are a number of comments for the authors to consider. As a general comment, the authors make a number of grandiose assertions as to the impact of the work in the context of improving plant productivity. As detailed below, there are flaws in some of these statements and if unchecked would add further fuel to the growing misconceptions in the literature that these enzymes have potential use in plant productivity enhancement. Their usefulness in such an application would require a highly efficient CCM (which they briefly touch on in the discussion) and that their sensitivity to oxygenation inhibition was not too detrimental (which was not assessed here).

The paper was indeed somewhat over-hyped in its assertions as a result of being sent in the previous round to *Nature*. We now made changes in the manuscript to appropriately tone down the statements and provide a more balanced discussion of the results. We feel this was a very correct criticism that make us much happier with the updated paper.

Here are some relatively minor comments the authors should consider/address:

ABSTRACT

1. What defines a catalytically superior Rubisco is highly context-dependent. In plants and algae, a Rubisco with a high CO₂/O₂ specificity would broadly be described as superior, somewhat independent of k_{cat}. So for the authors to say they are searching for "catalytically superior variants" is misplaced and at odds with what they actually measured for which is actually for Rubisco with higher (substrate saturated) rates of CO₂-fixation. (Notably, in nature the CO₂ fixation rates reach would unlikely be anywhere close to the saturated rates without a cyanobacteria equivalent CCM given the high K_mCO₂'s required).

The term "catalytically superior variants" was replaced with "fast carboxylating variants".

2. While 143 Rubisco genes were synthesized, only 78 were actually purified and biochemically tested.

We now have a collection of 105 purified and active enzymes. We now present the results of all of them and explain that some fraction (much smaller than before) were insoluble.

3. Not sure of the reasoning as to why these enzymes would be "especially promising for biotechnology" or for "feeding a growing population" (a rather grandiose, unsubstantiated statement)?

We apologize for the originally grandiose text. We removed this statement.

INTRODUCTION

4. No mention of the Rubisco oxygenation reaction is mentioned. The focus is primarily on Rubisco being a "slow" as its only kinetic constraint. For the novice reader, it would be unclear as to the impact O₂ inhibition influences catalysis - one of the main reasons it is such an intensively studied enzyme. and a major constraint on Rubisco catalytic efficiency (its not just a "slow" enzyme).

Oxygenation is indeed an important aspect of rubisco's kinetics. Since the initial submission of the manuscript, we have measured the specificity factor ($S_{C/O}$) for our fastest 7 candidates and included it in the results section. Further and given the referee's comment, we now also elaborate on the specificity-rate tradeoff of rubisco.

5. P2 - replace "made relatively little progress" with "made limited progress". It might be worth mentioning that (real) successes in improving the net carboxylation capacity of Rubisco have only been achieved for Form I and III isoforms (the latter by Wilson et al., 2016, Scientific Report) and only by directed evolution. (See also the recent paper by Yu and Whitney, 2019, Int J Mol Sci 20: 5019)

The wording was updated according to the referee's comment. We also appreciate the pointers provided by the referee and have now included them in the manuscript.

6. The authors should again be cautious in their terminology. In the last paragraph on P2, to be more accurate the text should state "in search of catalytic outliers in terms of CO₂ fixation speed and CO₂ affinity". Similarly, what defines "superior CO₂ fixation kinetics" is arbitrary. For example, according to the data in table 2 the carboxylation efficiency of the enzymes analyzed in this study are all 3-10 fold lower than plant Form I Rubisco and so one could say none are "superior". Certainly one would anticipate the CO₂/O₂ specificity of eukaryotic sources of Form I Rubiscos would similarly be "superior" to those measured in this study.

In retrospect we fully agree with the referee and updated the wording as well as added: "... in terms of CO₂ fixation speed (i.e., k_{cat} for carboxylation)." to the text.

RESULTS

7. Does the assertion on P3 that additional metagenomic unveiling of Rubisco sequence diversity will diminish the dominance of Form I Rubisco sequence diversity take into account the sequence diversity (and multi-gene copies) of RbcS?

The analysis regarding the sequence diversity of rubisco was based only on the large subunit. We agree that not accounting for the diversity of the small subunit can skew the degree of sequence diversity in Form I rubiscos compared to all other forms that do not have a small subunit. We now make it clear in the text that the analysis was based on the large subunit and changed the wording of the above sentence to:

"Notably, while 97% of the sequenced rubiscos are form-I, they account for only ≈50% of the sequence diversity of rubisco's large subunit. It is likely that future metagenomic efforts will uncover novel rubisco sequences that shall further expand the contribution of form-II, III and II/III rubiscos to the global sequence diversity of this enzyme."

8. P4; Suggest rewording to "folding and assembly of plant form-I rubisco" to align with the reference cited.

Done.

9. P4; "4 failed to express in *E. coli*". Please clarify if referring to there being no evidence of functional enzyme biogenesis (solubility), or no detection of soluble LSu made (which would suggest a cloning error).

All plasmids cloned for this study were sequence verified (by Sanger and/or Next-Generation sequencing). The term "failed to express" was used instead of the appropriate "insoluble". Accordingly, we changed the wording in the manuscript.

10. P4; replace "3-phosphoglycerate will be formed" with "2-phosphoglycolate will be formed".

The enzymatic coupled assay used here couples 3-phosphoglycerate production to oxidation of NADH. With that respect, the oxygenation reaction is problematic because 3-phosphoglycerate is also formed (alongside 2-phosphoglycolate). This is why if "3-phosphoglycerate will be formed via a promiscuous oxygenation reaction", the coupled assay would be biased. To make it more clear, we now write:

"A potential caveat to this assay is that if O₂ levels are sufficiently high, 3-phosphoglycerate will be formed via a promiscuous oxygenation reaction (alongside 2-phosphoglycolate), resulting in a deviation between the rates of NADH oxidation and carboxylation."

11. Not sure why Table 2 is in the SI, it is more suited to the main text and could easily replace Fig 2 that could be moved to the SI (but I would note it is a very clear and nicely presented figure).

This was the result of tight space constraint in the previous submission. We agree with the referee and have now moved Table S2 to the main text. As elaborated in the point below, we also included further columns to the table.

12. Additional columns should be included in Table 2 to include the following additional information. (a) The RbcL subunit stoichiometry of each enzyme. (b) the carboxylation efficiency values (i.e. kcat/KM) at 25°C. (c) the corresponding kcat derived by the spectrophotometric assay (for comparative assessment of accuracy, recognising they were done at different temps). (d) inclusion of comparative examples of literature values for plant Form I Rubisco (so that readers can clearly ascertain that while the form II and cyanobacteria enzymes may be faster, their low CO₂-affinities compromise their carboxylation efficiencies).

Following the referee's comment, (a) and (b) and (d) were added to the Table which is now in the main text. We further included the specificity-factor ($S_{c/o}$) of each variant, as we were able to determine them since the initial submission of the manuscript. The corresponding specific activities derived by the spectrophotometric assay is performed under a different temperature and level of CO₂ saturation and thus can be confusing when out of context - we give a comparison with the description of those issues in the SI. Importantly, we added the following paragraph to the main text:

“The CO₂:O₂ specificity factors (SC/O) of the 7 fast rubiscos were also measured to assess the relative capacities of these enzymes to catalyze carboxylation over oxygenation of ribulose 1,5-bisphosphate (see Methods). Compared to *S. elongatus*, which has a specificity factor of 42.7±2.8, the L₂ rubisco from *Gallionella sp.* is fourfold less specific (10.0±0.1). In general, the specificity of all our bacterial rubiscos was significantly lower than of *S. elongatus*, which is already more than twofold less specific than the median plant rubisco (plant variants exhibit a specificity factor range of ≈80-120 (Flamholz et al, 2019)). These measurements thus indicate that in order to support considerable carboxylation flux, the novel rubiscos would have to operate in the context of a carbon concentrating mechanism (CCM) or in otherwise high CO₂ environments.”

DISCUSSION

13. P8; arguably only tested 78, not 143, Rubisco variants were tested.

We now have activity values for 105 variants out of 143 in the initial library. The sentence was changed to:

“Here we tested >100 rubisco variants...”

14. It is stated that one basal clade of organisms tended to have higher kcats, however, it is impossible to ascribe in Figure 2 any correlation with the enzymes analysed in table 2.

Following the comment, we now annotate the bars in Figure 2 that correspond to the 7 variants that were further analyzed.

Also lacking is any description of the comparative amino acid similarity among the Form II enzymes characterized in Table 2? Is there sufficient similarity for the authors to start speculating about specific RbcL amino acids that influence kcat?

The clustering algorithm used to choose representative sequences in this study found variants that are less than 90% identical to one another at the amino acid level (which results in covering the sequence space at a resolution of 90% sequence-identity). This means that despite being in close proximity on the dendrogram in Figure 2, each variant is at least 40 amino-acids different than any other variant in the set (the length of rubisco is ≥400a.a).

Unfortunately, despite being an interesting question raised by the referee, the relatively large sequence distance alongside the small dynamic range in kinetic properties was insufficient to speculate/conclude anything compelling. That being said, we fully agree with the referee that looking at the relationship between sequence-similarity and kinetic-similarity is valuable for studying rubisco as well as other enzymes (and, in fact, we wrote about that in a recent review (Davidi et al. 2018)). In a follow-up work we are now looking into other members of the clusters of those fast variants. Hopefully, analysis at higher resolution would allow us to identify positions that are important for catalysis, that could be specifically targeted in future studies.

15. To make the assertion that expression of *Gallionella* Rubisco would not be problematic in organisms with a CCM is making the assumption their Sc/o is not too crippled. Based on prior publications from this group one might suspect the Sc/o of the high kcat enzymes might be particularly impaired. As modelled

by Sharwood (2017, New Phytologist), such enzymes would still perform more poorly in a C4 plant than that native plant Rubisco.

Having also measured $S_{C/O}$, we now know that these present relatively low affinity to CO_2 and $S_{C/O}$ values. Indeed, in the context of C_4 plants, that will not be sufficient to elevate fixation rates. That being said, tradeoffs in rubisco kinetics seem to be relaxed for variants that act in organisms with CCMs (Iñiguez et al. 2020). It is possible that the fast variants discovered here would be a promising starting point for directed evolution in CCM containing model systems.

16. Replace "them to target the fast" to "them integrate the fast"

Done.

17. The last paragraph needs to acknowledge the limitations of the study. This study only focused on one facet of Rubisco kinetics (CO_2 -fixation speed and affinity) - it did not examine the properties of its other two substrates (RuBP and O_2), both of which are needed to truly determining the relative "superiority" of a Rubisco in differing physiological contexts - in particular within C3 and C4 plants in which appears the underpinning objective based on the last sentence of the abstract.

As indicated by the referee, the fast variants discovered here exhibit limitations. Specifically we report relatively low affinity to CO_2 and $S_{C/O}$ values. Indeed, in the context of C_4 plants, that will not be sufficient to elevate fixation rates. On the other hand, and as pointed out in our response to point (15), tradeoffs in rubisco kinetics seem to be relaxed for variants that act in organisms with CCMs (Iñiguez et al. 2020). Our variants, therefore, may prove to be a promising starting point for directed evolution in CCM containing model systems. Upon the removal of several overhyped statements in the manuscript (including in the abstract), we believe that along with major edits (as can be seen in the main text), the last paragraph of the discussion now accurately reflects the findings in the presented study.

Referee #1 additional comments:

There are still some statements in the revised version that may need to be addressed.

The updated manuscript was revised to reflect a more balanced view of our findings. We hope that the referee will agree.

I would also point out Review 2 comments on Fig 5 are unjustified - they (and the authors here) don't seem to appreciate that assaying at saturating substrate levels is not critical when fitting a Michaelis-Menten fit. The critical component is ensuring accuracy around the K_M value as the initial slope of the fit actually represents the value for V_c/K_M (catalytic efficiency). So in essence, those who focus on getting V_c values using too-saturating [substrate]'s actually get lower V_c/K_M values which don't fit the experiment data.

According to the referee's comment we now updated the response to referee #2 and added the following paragraph to the results section:

"While the catalytic capabilities of enzymes are traditionally evaluated by their k_{cat} or k_{cat}/K_M values, it is important to consider the physiological conditions in which those enzymes are

active. For example, C_3 plants do not have a CCM and tend to have rubisco enzymes that are highly specific and relatively slow (Figure 5a). In C_4 plants, where CO_2 is somewhat concentrated, rubiscos are faster and less specific (Figure 5a). As the CCM efficiencies increase, k_{cat}/K_M values seem to decrease and k_{cat} values increase, as in the case of form II rubisco. Due to the high K_M values of form II rubiscos, biochemically determining their rate at full substrate saturation becomes technically highly challenging and may not actually reflect their true k_{cat} value. This is because such high concentrations of CO_2 can result in substrate inhibition, as well as affect the pH of the solution, to which the carboxylation reaction is sensitive. We were able to infer k_{cat} and K_M values for those enzymes by measuring the rate around the K_M values. The most robust inference is for the initial slope of the Michaelis-Menten (MM) curve which represents k_{cat}/K_M . Notably, the MM curve for *Gallionella* sp. is above that of *S. elongatus* in all tested CO_2 concentrations, implying that when oxygen is not limiting, rubisco from *Gallionella* sp. will catalyze more CO_2 fixation reactions compared to *S. elongatus* rubisco under any physiologically relevant CO_2 concentration. Plant rubiscos, however, will catalyze more CO_2 fixation reactions at CO_2 concentrations lower than $50\mu M$.

One other comment is the authors comments on the $k_{cat} - S_{CO_2}$ trade-off need to carefully consider this contention is largely outdated as being a constraint - and base around a very limited dataset. As the authors actually indicate, there is quite a bit of flexibility in this relationship among organisms with a CCM (e.g. C_4 -plants have 2-fold higher k_{cats} than C_3 -plant Rubisco but the same, or better, S_{CO_2} e.g. Maize, sorghum etc) and certainly any such trade-off does not hold for improved Rubiscos generated by directed evolution. Any evidence of the trade-off in aquatic organisms breaks down even more as their Rubisco have evolved in O_2 and CO_2 environments that vary by the hour depending where they find themselves in the water column, irrespective of whether or not they have a CCM.

We fully agree with the referee and feel that the current manuscript highlights the fact that the tradeoff hypothesis is based on a sparse set of rubiscos. Indeed, we write in the manuscript that:

“However, this perception is based on kinetic measurements that cover only a tiny fraction of rubisco’s natural diversity. In a recent analysis, we showed that measurements of rubisco homologs from a set of divergent organisms has called the robustness of the ‘optimality hypothesis’ into question (Flamholz et al, 2019).”

Further, in the discussion we now mention that:

“Indeed, tradeoffs in rubisco kinetics seem to be relaxed for variants that act in organisms with CCMs (Iñiguez et al, 2020).”

Referee #2:

Overall, this is a very solid study that focuses on screening the kinetic diversity of RubisCO's carboxylation reaction across the phylogenetic tree. The study extends on earlier efforts in the field, in particular by the Tabita, Perner and Kerfeld labs. Compared to the reports from the Tabita (several publications over the years) and Perner labs (ISME Journal 2019), this study comprises a bigger data set. Its exclusive focus on existing RubisCOs makes it different from the Kerfeld study (Nat. Comm. 2017) that additionally took evolutionary considerations into account (through assessing the kinetic diversity of existing and extent RubisCOs by ancestral reconstruction from phylogenetic trees). In summary, the manuscript by Davidi et al. complements very nicely these other publications.

The experiments are thoroughly designed and the data is of high quality. The study has its merits for (more) systematically extending the set of kinetic data for RubisCO. However, the study does not find any correlation patterns between the kinetic parameters of individual RubisCO homologs and phylogenetic or environmental signatures, which would have been very exciting and a major advancement for the field (note that the Kerfeld study did identify some trends).

The authors claim that they were able to identify a RubisCO homologue that is two times faster than any other RubisCO homolog reported before. First of all, the Tabita lab reported already 2006 on a type III RubisCO with 23 s⁻¹ (although at elevated temperature). Second of all, such findings are not uncommon for (medium sized) phylogenetic screens (see related work from enzyme screening labs).

The Tabita lab indeed found a form-III rubisco that catalyze 23 s⁻¹ from the hyperthermophiles archaea *A. fulgidus*. We now cite this paper explicitly in the text. However, this value was measured at 83°C. At 25°C, this enzyme was marginally active in our hands, and indeed, if we correct the reported value using $Q_{10}=2.2$ (Cen & Sage, 2005), the k_{cat} at 25°C would be ≈ 0.75 s⁻¹.

Also note that the reported CO₂-fixation activity is still almost five-fold lower than that of other CO₂-fixing enzymes (especially PEP carboxylase and crotonyl-CoA carboxylase/reductase). Overall, these findings are not unexpected and do not essentially change our understanding of RubisCO and its catalytic limitations compared to other CO₂ fixing enzymes. There are also some experimental doubts on the reported "fastest RubisCO" (see below, detailed comments).

We fully agree with the referee that other carboxylases have significantly faster carboxylation rates compared to rubisco. Part of it may be due to the fact that rubisco uses CO₂ as a substrate instead of bicarbonate. In the past, we and others suggested using PEP carboxylase as part of a synthetic CO₂ fixation pathway. Such efforts are being pursued by several labs including Tobias Erb's lab at the Max Planck institute. Unfortunately though, making PEP carboxylase work *in-vivo* in an autocatalytic flux-mode for CO₂ fixation is highly challenging. Form II rubiscos, on the other hand, have been shown to support CO₂ fixation flux when heterologously expressed in autotrophic organisms as well as in synthetic systems in heterotrophic microorganisms (e.g. Gleizer et al., *Cell*, 2019). It is indeed an invaluable effort of the community to try other carboxylases for CO₂ fixation *in-vivo*, yet we believe that synthetic rubisco-based CO₂ fixation in living organisms is still a highly valuable avenue to try at this point in time.

Most importantly, the study fails in respect to a crucial point. Note that one important kinetic parameter in RubisCO is carboxylation activity. An equally important parameter, however, is the specificity of

RubisCO to discriminate between CO₂ and O₂. It has been reported several times that RubisCO shows an inverse correlation ('trade-off') between carboxylation activity and specificity. The faster a RubisCO homolog, the lower is typically its specific reaction with CO₂. This study does not report on a single RubisCO that apparently breaks this inverse correlation, which would have been challenging the longstanding dogma in the field and be of immediate relevance for any efforts to improve photosynthetic yield in plants. In other words, the two-fold faster RubisCO is not really useful, if it is not also more specific for CO₂. Finding a faster and more specific RubisCO is a holy grail, but this study did not provide such a long-sought-after enzyme.

Having now measured the specificity factor of our top 7 variants as we report in the updated text and table, we now know that their CO₂:O₂ discrimination capacity is low but still in the ballpark of efficient carboxylation kinetics at physiologically relevant CO₂ concentrations in organisms with CCM.

As the referee pointed out, finding rubisco variants that break the tradeoff line would represent a holy grail and utmost exciting news. While we did not find such rubiscos in this study, we believe that the presented approach is a valuable path toward discovering such variants if they exist, especially in light of our discoveries of the highest k_{cat} carboxylating rubiscos to date at 25° C. Specifically we view the bioinformatic pipeline as well as the plate-based carboxylation assay as useful contributions to the field as they allow to systematically explore the kinetic space of rubisco at unprecedented throughput.

In summary, this study is a very valuable contribution to the field of RubisCO biochemistry and a step forward, but rather an incremental one.

More detailed comments on the manuscript:

Abstract: Please remove the statement that the identified enzymes will contribute to the challenge of feeding a growing world population. This is too far-fetched and the enzymes identified show extremely high K_m values for CO₂, which would be a huge challenge for any efforts of implementing them into plants.

This statement was removed from the text.

Introduction: "It has been suggested that the catalytic properties of this enzyme may already be optimized and thus cannot be straightforwardly improved." This is a misleading statement because this argument is only valid if one considers oxygenation versus carboxylation, which is in fact not considered in this study.

We have now added the following sentence to the introduction:

“Specifically, rubisco can use O₂ as a substrate instead of CO₂ to catalyze an oxygenation reaction. A tradeoff between greater CO₂:O₂ specificity ($S_{C/O}$) and a greater carboxylation rate was previously reported (Tcherkez et al, 2006; Savir et al, 2010). In this tradeoff, the enzyme might have reached what was discussed as a fitness optimum.”

Further, as indicated above, since the initial submission of the manuscript, $S_{C/O}$ was determined for the fastest 7 variants (reported in the updated results section).

When only looking at carboxylation activity in anaerobic or carbon concentrating mechanisms, there had been RubisCOs reported with 14 s⁻¹ (*Synechococcus*) and 23 s⁻¹ (*Archaeoglobus*), which clearly indicates that there is less constraints, if the oxygen side reaction can be neglected.

As was highlighted by the referee, organisms that live in micro-aerobic or anaerobic conditions can express rubisco homologs that are less kinetically constrained, as the oxygenation reaction can be neglected. Indeed, we state in the manuscript that among the fastest 7 variants, at least 5 live in low to no oxygen habitats.

Regarding other fast rubiscos that were previously reported: our understanding of the 23 s⁻¹ value is detailed above. As for the 14 s⁻¹ value reported for *S. elongatus*, it is markedly higher compared to 5 other papers that measured the rate of that same variant (either from *Synechococcus* PCC6301 or PCC7942, which encode for an identical enzyme). In fact, the second fastest value reported for *S. elongatus* is 11.8s⁻¹ and the median k_{cat} value for this enzyme is 11.6 s⁻¹.

In general, oftentimes we see remarkable variability across kinetic measurements between different papers (see more discussion e.g. in (Davidi et al. 2016)). To manage those variations we referred to the median k_{cat} value across all measurements, and at the same time, included rubisco from *S. elongatus* as a control in our isotopic labeling experiments.

Lastly, a k_{cat} value of 14 s⁻¹ would still be about 40% lower than that of the fastest rubisco discovered in this study (22 s⁻¹).

Figure 5: The data on the *Gallionella* enzyme seems to be very sensitive to the fitting, as the enzyme was not measured under fully saturating conditions. It rather looks like the enzyme would level off at 16 s⁻¹, which would be much closer to the 14s⁻¹ reported for the *Synechococcus* enzyme, but this remains unclear because the data at higher concentrations was not collected (and not experimentally compared to the *Synechococcus* enzyme as reference).

Going over dozens of kinetic reports of rubisco, the concentration of CO₂ used for kinetic measurements is up to 40 mM of NaHCO₃, which would correspond to about 400 μM of CO₂ (at pH 8 and 25° C). This is also true for the reported studies for *S. elongatus*, which, on average report a median k_{cat} of 11.6 s⁻¹. Having a K_M value of about 200μM, determining the k_{cat} for *S. elongatus* at 400μM of CO₂ is more sensitive to fitting than was done in our study (500 μM of CO₂). Accordingly, at 500 μM CO₂, the specific activity of *S. elongatus* would be about 8 s⁻¹ compared to about 15 s⁻¹ for the *Gallionella* enzyme. It is important to note that higher concentrations of CO₂ are not easy to test, since going above 500 μM of CO₂ can result in substrate inhibition, as well as affect the pH of the solution, to which the carboxylation reaction of rubisco is sensitive. It is likely that rubisco studies have traditionally been using relatively low concentrations of CO₂ since plant rubiscos have been mostly assayed and their K_M values are on the order of 50 μM. In general, in most if not all studies the k_{cat} is extrapolated from the data in a similar manner. And yet, as also indicated by referee #1, when fitting a Michaelis Menten curve, it is most important to sample around the KM value, as the initial slope of the curve represents k_{cat}/K_M .

Accordingly, we now added the following paragraph to the results section:

“While the catalytic capabilities of enzymes are traditionally evaluated by their k_{cat} or $k_{\text{cat}}/K_{\text{M}}$ values, it is important to consider the physiological conditions in which those enzymes are active. For example, C_3 plants do not have a CCM and tend to have rubisco enzymes that are highly specific and relatively slow (Figure 5a). In C_4 plants, where CO_2 is somewhat concentrated, rubiscos are faster and less specific (Figure 5a). As the CCM efficiencies increase, $k_{\text{cat}}/K_{\text{M}}$ values seem to decrease and k_{cat} values increase, as in the case of form II rubisco. Due to the high K_{M} values of form II rubiscos, biochemically determining their rate at full substrate saturation becomes technically highly challenging and may not actually reflect their true k_{cat} value. This is because such high concentrations of CO_2 can result in substrate inhibition, as well as affect the pH of the solution, to which the carboxylation reaction is sensitive. We were able to infer k_{cat} and K_{M} values for those enzymes by measuring the rate around the K_{M} values. The most robust inference is for the initial slope of the Michaelis-Menten (MM) curve which represents $k_{\text{cat}}/K_{\text{M}}$. Notably, the MM curve for *Gallionella sp.* is above that of *S. elongatus* in all tested CO_2 concentrations, implying that when oxygen is not limiting, rubisco from *Gallionella sp.* will catalyze more CO_2 fixation reactions compared to *S. elongatus* rubisco under any physiologically relevant CO_2 concentration. Plant rubiscos, however, will catalyze more CO_2 fixation reactions at CO_2 concentrations lower than $50\mu\text{M}$.“

Thank you for submitting a revised version of your manuscript. Your study has now been seen by both original referees, who find that their main concerns have been addressed and are now in favour of publication of the manuscript. There now remain only a couple of minor editorial issues that have to be addressed before I can extend formal acceptance of the manuscript:

1. Reviewer #1 has offered several suggestions regarding the style and clarity of the text. Please implement at your discretion.
2. Please consider changing the title according to the suggestion of reviewer #2 - I agree that this wording would be better in long-term perspective,
3. Please check the comments from our data editor Vivian Killet in the figure legends section and add the requested information (the file is attached).
4. Please submit up to five keywords.
5. Please add Author Contributions and Conflict of Interest sections before the References section.
6. Figure panel 3b and Appendix Figure S5 are not referred to in the manuscript text.
7. Please remove figures from the main manuscript text and place figure legends after the References section.
8. Please place Table 1 and Table 1 legend at the end of your manuscript after your Figure Legends Section
9. Table S1 should be turned into Table EV1 with a legend inserted into the xlsx file in a separate tab.
10. Table within the appendix file has no nomenclature. Please rename to Table S1 and update callouts accordingly.
11. Please remove all 'Available At' from the References section.
12. Please add a short table of contents at the beginning of the Appendix file.
13. Papers published in The EMBO Journal are accompanied online by a 'Synopsis' to enhance discoverability of the manuscript. It consists of A) a short (1-2 sentences) summary of the findings and their significance, B) 2-3 bullet points highlighting key results and C) a synopsis image that is 550x300-600 pixels large (width x height, jpeg or png format). You can either show a model or key data in the synopsis image. Please note that the size is rather small and that text needs to be readable at the final size. Please send us this information along with the revised manuscript.

Please let me know if you have any further questions regarding any of these points. You can use the link below to upload the revised files.

Referee #1:

To recap the core advances of the paper; it describes a comprehensive bioinformatic mining exercise of available Rubisco RbcL sequence, their phylogenetic arrangement into clusters, the development a higher throughput (not necessarily a high throughput) kcat assay that allowed a broad survey of representatives from the differing Form II and Form II/III Rubisco clusters that revealed high kcat variants. The kinetics for 7 of these enzymes were more comprehensively examined in detail to reveal a greater catalytic diversity than previously perceived among the Rubisco superfamily - in particular the increased extension in the achievable kcat range. While I continue to find this work impressive both experimentally and intellectually, there remain a number of points to clarify and correct - as detailed below. Importantly I do not see a requirement for further experimentation. The deficiencies primarily relate to providing details on what biotech application(s) the authors feel their data will truly benefit and a better appraisal of the CCM literature - in particular how these might integrate alternative oligomeric Form II Rubisco isoforms. Currently the CCMs they refer to comprise Form I Rubisco where the RbcS are required for structural CCM integrity - they have overlooked consideration of the CCMs that encase Form II Rubisco and whose structural componentry should be a target for future investigation.

1. Title; Replace "by" with "following". The paper does not describe how sequence diversity was used to identify differences in kcat.
2. Abstract; Replace "relatively slow rate of plant rubiscos led to effort that made" with "poor carboxylation properties of plant rubisco have led to effort that make". As recent cross scaling modelling in crops has confirmed, the desired kinetic improvements to Rubisco are >20% increases in specificity and >30% increases in carboxylation efficiency - increasing kcat alone will do nothing in C3-crops.
3. Abstract; related to point 1 above - sequence mining does not inform Rubisco kinetics so the 3rd sentence needs to be reworded to something like "Here we systematically mined genomic and metagenomic data for ~33,000 unique Rubisco large subunit (RbcL) sequences which clustered into ~1000 similarity groups.
4. Abstract; the last sentence should be deleted or more clearly explained (along with the last sentence of the discussion). As indicated in points 1 and 3, the current wording infers the genomic mining process of the pipeline can be used to identify kinetic variability. This is not the case. The experimental pipeline is also limited to assaying enzymes where catalytic activity can be coupled to NADH oxidation (and likely NAD reduction). So what other "enzymes of interest" or "biocatalysts" does this include? This level of detail is not expanded upon in the discussion. I use the phrase again - such grandiose statements need to be accompanied by justifiable examples or simply excluded.
5. P1, as indicated by the full name for rubisco - it only comprises enzymes that catalyse RuBP.

Thus there are really only 3, not 4, distinct forms of Rubisco. The literature is tarnished by this inaccuracy and the authors should adjust their text to indicate that "there is a family of structurally related rubisco-like proteins that are denoted Form IV rubisco despite not catalysing RuBP, CO₂ or O₂"

6. P2, replace "across all known enzymes" with "across the catalytically characterised enzyme range". As the authors indicate, of all the "known Rubiscos", only a small percentage have been kinetically characterised.

7. With regard to the sentence beginning "If rubisco's carboxylation kinetics are not..." greater consideration needs to be made in providing the reader with a better appreciation around what known examples there are already in nature with regard to evolved faster rubiscos. The classic examples are the CCM correlated up to 2-fold differences in *k_{cat}* among closely related C₃ and C₄-plant Rubisco and the even higher *k_{cat}* of cyanobacteria Rubisco where its encapsulation with carboxysomes ensure CO₂ saturation (noting the assertion O₂ is excluded from these structures is circumstantial and yet to be biochemically demonstrated). This suggests further kinetic exploration of Rubisco from origins where low O₂ pressures are experienced might reveal other fast carboxylating forms of the enzyme.

8. P3, Reword the 3rd and 4th sentence in the results to "...a phylogenetic tree and the rubisco-like enzymes removed as they lack carboxylation activity leaving ~33,000 non-redundant variants. While rubisco isoforms that do not share..."

9. P3, last sentence of first paragraph "... as the sequence-space of wild RbcL sequences". As Form I Rubisco data is included it is important to again clarify the analysis is limited to RbcL.

10. P4, section heading should be "Expanding the space of *k_{cat}* characterised rubiscos by 4-fold" as only this parameter is characterised in the catalytic survey.

11. P4, Delete the sentence beginning "The assembly process is not..." as it is simply not true, and what "assembly intermediary" might be present they are in finite levels and do not bind CABP tightly or at all. I would suggest the authors justify their exclusion of undertaking a Form I Rubisco survey on the basis that "chaperone compatibility issues between even highly similar plant rubiscos make it infeasible" As shown in Aigner 2017, tobacco Rubisco biogenesis was poorly compatible with the Arabidopsis Rubisco ancillary proteins and Sharwood et al 2016 (Nature Plants) demonstrates how the assembly requirements of monocot Rubisco are not met in dicots (tobacco).

12. P4, Why exactly is Form-II a promising group for identifying biotechnology useful variants? What parameters are deemed useful? What biotech applications are being referred to? I would suggest deleting this unnecessary and ambiguous sentence.

13. P5, 2nd paragraph - requires a complete overhaul. I would suggest it reads as follows:

"To carboxylation rates of each purified Rubisco were determined using a spectrophotometric NADH coupled assay where 3-phosphoglycerate (3-PGA) production is coupled to NADH oxidation (Fig 3a, Kubie, Lily & Walker). In our assay two 3-PGA are produced, and hence 2 NADH oxidised, for each RuBP carboxylated by rubisco (see methods). A potential caveat to the assay is if RuBP oxygenation occurs where a 3-PGA and 2-phosphoglycolate are produced resulting in an underestimation of the rubisco substrate saturated carboxylation rate. To minimise this issue the spectrophotometric assays were undertaken in assays equilibrated in a nitrogen atmosphere comprising 4% (v/v) CO₂ and 0.2% (v/v) O₂ in a gas-controlled plate reader at 30°C. These conditions ensured the rubisco is fully carbamylated (CO₂-activated) before adding RuBP to initiate catalysis, the assay CO₂ levels were saturating and the occurrence of RuBP oxygenation trivial. The last sentence of the current paragraph should be deleted unless the authors more fully demonstrated in the methods/supplementary section what the dissolved [CO₂] and [O₂] in the assays are under these gas conditions and how these correspond to the most extreme *K_m* values of [O₂] and [CO₂] by other form II and prokaryotic Form I Rubisco.

14. P5, 3rd paragraph. Would suggest simplifying text to "we quantified the rubisco catalytic site content in each assay allowing the determination of carboxylation turnover rate per active site (s-

1).

15. P5 last sentence of paragraph 5 is confusing. Quantifying Rubisco activities in crude lysates is faster than having to purify the enzyme - such is the beauty of the tight and specific binding properties of CABP to Rubisco. In fact, one can argue a deficiency of the current NADH-assay used is that there appears an apparent need to purify Rubisco (slowing throughput dramatically). One assumes purification was undertaken to eliminate NADH oxidation reactions by *E. coli* proteins causing too much noise that would influence the *k_{cat}* measures of the poorly expressing form II Rubiscos. So in essence, the Rubisco screening assays employed in the two references provided were problematic as they did not undertake the CABP quantification approach undertaken in this paper to allow kinetic differences to be distinguished from variation in rubisco solubility. As for the activation kinetics, one assumes the NaHCO₃ levels in their assays were suitably saturating.

16. P6, paragraph at top of page is superfluous and can be deleted

17. P6, last paragraph. To be more specific suggest rewording to "The carboxylation *k_{cat}* and *K_M* properties for a subset of 7 promising Rubisco variants were undertaken using high precision ¹⁴CO₂-fixation assays. These assays were undertaken....." Use of emotive terminology like "gold standard" is unnecessary - and non-scientific.

18. P7, at top of page - rewrite as "(Figure 5b) wit the fastest rubisco from a soil member of the Gallionella genus (OGS68397.1) with a *k_{cat}* of 22.0 +/- 0.1 s⁻¹."

19. P7, same paragraph - replace awkward text with "We note the form-III rubisco from the hyperthermophilic archaea *Archaeoglobulus fulgidus* has a *k_{cat}* of 23 s⁻¹ at 83oC (Kreel 2015) however we found the enzyme shows very little activity at 25oC, consistent with a *k_{cat}* of ~0.75 s⁻¹ predicted assuming a Q₁₀ of 2.2 (Chen & Sage)."

20. P7, 2nd paragraph - again replace awkward text with "were also measured to compare the rates of RuBP carboxylation over oxygenation under equal [CO₂] and [O₂].....In all cases the specificity of all bacterial rubiscos were significantly lower than *S. elongatus* Rubisco, which is...."

21. P8, 2nd paragraph. This newly written paragraph lacks accuracy and clarity. The 2nd and 3rd sentences are incorrect, the specificities of C₃ and C₄ Rubisco are pretty much identical - the authors should look at the modern specificity measures recorded in their Flamholz publication. What differs is the *k_{cat}* and *K_M* for CO₂. I am unsure of the basis for the comment *k_{cat}*/*K_M* (carboxylation efficiency) changes with CCM efficiency. For example, the carboxylation efficiency of C₃-plant and C₄-plant Rubisco are highly comparable. This is shown in Sharwood 2017 (New Phytologist) where it is demonstrated putting a C₄ Rubisco into a C₃ plant will have little, mostly no, improvement to photosynthesis. Indeed this is even shown in figure 5a where the initial slopes of the C₃ and C₄ curves overlap.

I think what the authors need to focus on in this paragraph is that the carboxylation efficiency derived from the *k_{cat}* and *K_M* values mesured in vitro are used to map the operational *k_{cat}* in vivo which will vary depending on CO₂ environment experience by the enzyme, which is non-saturating if the organism does not possess a CCM or capacity to grow under low O₂ or anaerobic conditions.

22. P8, Figure 5 - more descriptive and simpler legend needed (remove methodology context as unnecessary). Suggest simplify to "Comparing the carboxylation *k_{cat}* and *K_M* of Gallionella sp Rubisco". Need to clearly indicate the Y- and X-axis are on a log scale. In panel b the Gallionella sp data point needs to be red. Should consider highlight *S. elongatus* data point in different color.

23. P9 - 2nd paragraph. As Archaea Rubisco has had no need to adapt to expression in *E. coli* I would suggest rewording the 5th sentence to "The assembly requirements of Archaea Form III rubisco from are not well met by *E. coli* and thus excluded from this study."

24. P9 - why do the authors consider exploring the sequence space of form III a worthwhile effort? Is it because of their alternative biological function which may have allowed for alternative evolutionary adaptation in terms of their CO₂ and O₂ fixation properties? If so, then articulate this as the rationale for their further study importance.

25. P9 - last paragraph. As indicated in opening paragraph, the principal of CCM Rubisco

recruitment are becoming clearer in the 3 paper cited - all involve form I Rubisco, all implicate the RbcS in having a structural role. How is this relevant to form II rubisco? Should we in fact be resolving the structural features and functionality of dinoflagellate CCM's which house form II Rubisco. A little bit more thought, context and detail is required here to explain the potential use of their data in a biological/biotech context.

26. P10 - last sentence of Discussion - a bit of an oversell. The in vitro testing capabilities developed in this paper have limited application. Essentially it is restricted to enzymes (biocatalysts) assayable using an NADH-coupled assay and limited by the enzyme E. coli expression, purification and quantification capacity. One could argue the assay is really only high-throughput comparatively to the traditional Rubisco assay methods. The compelling take home message is this paper demonstrates the potential of using a data mining approach to identify clusters of related enzymes from which representatives can be chosen to lessen the number of (or more strategically inform which) variants to kinetically evaluate - providing the necessary structure-function appreciation needed to bioengineer the enzyme.

27. P12 - given the buffer conditions used in the $^{14}\text{CO}_2$ assays the text on P8 is unsubstantiated. Firstly, high CO_2 concentrations do not inhibit Rubisco activity but high HCO_3^- ions can. Secondly, the pH of the buffer used will not be affected by 0.1M NaHCO_3 concentrations (two -fold the maximum used - something the authors can easily confirm with a pH electrode)

On a finishing note, the figures are really well designed and very clear (in both main text and supplementary data).

Referee #2:

The manuscript has (further) improved. I thank the authors for providing more data to support their work (which was actually not necessary, but is well appreciated), as well as modulating their language. I can only congratulate the authors for their work.

The only suggestion I might have is that the title could be adopted to better reflect the article's (excellent) findings. Claiming "faster", "higher", "better" is not very satisfying and might not withstand future findings and/or be misleading (e.g. if new RubisCOs of even higher activity are to be found). Maybe "Highly active RubisCOs discovered by systematic interrogation of natural sequence diversity"? might be a better suited title and put emphasis on the systematic analysis of many RubisCOs in parallel, which is in my point of view the key finding of the manuscript. But this might be more a consideration for the editorial office.

Authors performed the requested changes.

Thank you for addressing the final minor issues in the revised manuscript. I am now pleased to inform you that your manuscript has been accepted for publication.

Corresponding Author Name: Ron Milo

Manuscript Number: EMBOJ-2019-104081